# Land cover and management effects of ecosystem resistance to drought stress

Chenwei Xiao[1], Sönke Zaehle[2], Hui Yang[1], Jean-Pierre Wigneron[3], Christiane Schmullius[4], Ana Bastos[1]

[1]Department for Biogeochemical Integration, Max Planck Institute for Biogeochemistry, Jena, Germany
[2]Department of Biogeochemical Signals, Max Planck Institute for Biogeochemistry, Jena, Germany
[3]INRAE, UMR1391 ISPA, Université de Bordeaux, F-33140 Villenave d'Ornon, France
[4]Department for Earth Observation, Friedrich Schiller University Jena, Jena, Germany

*Correspondence to*: Chenwei Xiao (cxiao@bgc-jena.mpg.de)

**Abstract.** Drought events are projected to become more severe and frequent across many regions in the future, but their impacts will likely differ among ecosystems depending on their ability to maintain functioning during droughts, i.e., ecosystem resistance. Plant species have diverse strategies to cope with drought. As a result, divergent responses of different vegetation types for similar levels of drought severity have been observed. It remains unclear whether such divergence can be explained by different drought duration, co-occurring compounding effects, e.g., heat stress or memory effects, management practices, etc.

Here, we provide a global synthesis of vegetation resistance to drought and heat using different proxies for vegetation condition, namely the Vegetation Optical Depth (SMOS L-VOD) data from ESA's Soil Moisture and Ocean Salinity (SMOS) passive L-band microwave mission and EVI and kNDVI from NASA MODIS. Due to its longer wavelength, L-VOD has the advantage over more commonly used vegetation indices (such as kNDVI, EVI) in that it provides different information on vegetation structure and biomass and suffers from less saturation over dense forests. We apply a linear model accounting for drought and temperature effects to characterize ecosystem resistance by their sensitivity to drought duration and temperature anomalies. We analyze how ecosystem resistance varies with land cover across the globe and investigate the potential effects of forest management and crop irrigation. We compare estimates of ecosystem resistance to drought and heat as retrieved from L-VOD, kNDVI and EVI products.

We find that regions with higher forest fraction show stronger ecosystem resistance to extreme droughts than cropland for all three vegetation proxies. L-VOD indicates that primary forests tend to be more resistant to drought events than secondary forests when controlling for the differences in background climate, but this cannot be detected in EVI and kNDVI. The difference is possibly related to EVI and kNDVI saturation in dense forests. In tropical primary evergreen broadleaf forests, old-growth trees tend to be more resistant to drought than young trees from L-VOD and kNDVI. Irrigation increases the drought resistance of cropland substantially. Forest harvest decreases the drought resistance of forests. Our results suggest

that ecosystem resistance can be better monitored using L-VOD in dense forests and highlight the role of forest cover, forest management and irrigation in determining ecosystem resistance to droughts.

## 1 Introduction

Heat waves and drought events have become more frequent since the last century and this trend is expected to continue across many regions under projected environmental changes with high confidence (IPCC, 2021). These events disturb ecosystems and can potentially weaken the land carbon sink (Schwalm et al., 2012; Reichstein et al., 2013; Zhang et al., 2019). During the past decade, the negative effects of climate variability and change contribute to a decline in the land sink in tropical forests across Amazonia and counterbalanced the $CO_2$ effects in many regions (Friedlingstein et al., 2022). However, drought impacts differ among ecosystems depending on the ability of ecosystems to maintain their functioning during adverse conditions, i.e., the ecosystem resistance (Gessler et al., 2020; Ingrisch and Bahn, 2018) and recovery trajectory following the disturbance (Schwalm et al., 2017; Wigneron et al., 2020; Wu et al., 2022; Yao et al., 2023). Ecosystem resistance is defined as the concurrent impact of a disturbance on response parameters. However, event-specific resistance may differ under different drought conditions. We applied a long-term resistance definition as the vegetation sensitivity to drought duration over multiple years, making it consistent for spatial comparison. Therefore, gaining a thorough understanding of the global patterns of ecosystem resistance and identifying the potential drivers behind their spatial variations is crucial for comprehending the impact of drought events on the land carbon sink. Potential drivers include climate background, plant species, biodiversity, tree height and ages and land use and land management. Land use and land management can potentially affect ecosystem resistance through changes in species composition, biodiversity and stand structure.

Ecosystem resistance to drought stress may vary with ecosystem composition. Experimental and site-based studies have compared the growth decline and mortality rate between different plant species and functional types during drought events. Gymnosperms show higher hydraulic safety margins than angiosperms, suggesting a higher tolerance to drought stress (DeSoto et al., 2020). On the other hand, higher biodiversity has been found to strengthen ecosystem resistance to drought events in temperate beech and thermophilous deciduous forests in drought-prone environments (Grossiord et al., 2014) and in grassland with stabilized ecosystem productivity (Isbell et al., 2015). However, previous studies provide only regional results. Furthermore, inconsistency in definitions of ecosystem resistance, considered drought duration (Slette et al., 2019), measurement time and frequency between plots, make the samples available for comparison limited.

To achieve a broader coverage and make consistent comparisons, satellite products (e.g., EVI, NDVI, VOD) and dynamic global vegetation models (DGVMs) are used to evaluate the ecosystem response during drought extreme events. With NDVI, Liu et al. (2022) have found an enhanced drought resistance from tree species diversity in dry forests such as xeric

woodland, subtropical dry forests and Mediterranean forests. Taller tropical forests have been shown to be more vulnerable to vapor pressure deficit (VPD) fluctuations during drought periods because of smaller xylem-transport safety margins, in studies based on Ku-band VOD (Liu et al., 2021) and SIF (Giardina et al., 2018). However, the opposite has been found when analyzing resistance to precipitation variability (Giardina et al., 2018) and terrestrial water storage anomalies (Liu et al., 2021), with taller trees being associated with higher resistance to soil-moisture deficit, based on the same vegetation proxies. However, some satellite products are less capable of detecting vegetation dynamics in dense forests. NDVI and EVI show saturation in dense forests (Li et al., 2021). kNDVI is better correlated with key vegetation parameters, such as leaf area index (LAI), gross primary productivity (GPP), and sun-induced chlorophyll fluorescence (SIF) compared to NDVI. It shows a higher resistance to saturation (Camps-Valls et al., 2021), but the signal still reflects mostly the upper canopy layer and cannot detect above-ground biomass changes within the vegetational volume, which represents the amount of carbon stored in above-ground vegetation. For complex ecosystems with a multi-layer structure, e.g., the Amazon rainforest, the impacts cannot be detected from observations of the top canopy greenness (Walther et al., 2019). Several studies have investigated the sensitivity of GPP to drought events (Bastos et al., 2020; Flach et al., 2018; Zscheischler et al., 2014). For example in Europe, both DGVMs and upscaled FLUXCOM GPP have suggested that GPP anomalies under low soil moisture conditions are less negative, or even positive, in pixels with more than 80% forest cover than in those pixels with lower forest cover. On the contrary, pixels with higher crop cover show stronger negative impacts on GPP anomalies compared to the pixels with low crop coverage in drought and heat events that occurred in 2003, 2010 and 2018 in Europe (Bastos et al., 2020). Over the globe, FLUXCOM GPP shows a reduced sensitivity to depleted soil moisture with increased tree cover (Walther et al., 2019). However, ecosystem fluxes are not directly observable at a regional scale beyond the footprint of flux towers (a few hundred meters). Above-ground vegetation biomass changes, on the other hand, can be retrieved globally from satellite data, being therefore useful to quantify the ecosystem resistance worldwide (Araza et al., 2023).

The majority of the land surface is managed by humans. By changing the biophysical and biogeochemical properties of the land surface and the plant functional traits, management practices also affect ecosystem resilience to climate extremes. For example, based on a stand-alone forest gap model, modifying forest density and structure by high-intensity overstory removal in conifer-broadleaf mixed forests in Central Europe considerably increased their growth resilience to droughts and decreased drought-induced mortality by two-thirds (Zamora-Pereira et al., 2021). For cropland, irrigation has been proven to be an effective strategy to mitigate the impacts of heat waves and drought events (Jia et al., 2019). Aside from directly alleviating soil water deficits and mitigating drought impacts, irrigation of land causes a global increase in evaporation of 32500 m$^3$ s$^{-1}$ (Sherwood et al., 2018) and a decrease in mean surface daytime temperature (Mueller et al., 2016). Such cooling can locally mitigate the effect of heat waves. The dependence of ecosystem resilience on tree species, height, size, age and land cover types also suggests that land management related to the above parameters may strongly affect the ecosystem response to extreme events (Condit et al., 1995; Nepstad et al., 2007; McDowell et al., 2008; McDowell and

Allen, 2015; Liu et al., 2021a, b). Nevertheless, there is a lack of studies linking forest management to ecosystem resistance

at a global scale. With the projected increased intensity and frequency of droughts and heat extremes in the coming decades, it is important to evaluate the role of various land management practices on the resistance and resilience of ecosystems to those events.

Aiming at a global analysis of ecosystem resistance based on ecosystem state variables, we use the Vegetation Optical

Depth (L-VOD) product based on L-band microwave emission observations from ESA's Soil Moisture and Ocean Salinity (SMOS) satellite, which can be related to aboveground biomass at annual timescales (Brandt et al., 2018; Fan et al., 2019; Qin et al., 2021). For comparison, we use EVI and kNDVI. Specifically, we use global L-VOD spanning from 2010 to 2020 to investigate the spatial variability of ecosystem resistance to heat and drought events. Ultimately, we explore several possible effects of land cover and land management, including dominant vegetation cover, forest fraction, irrigation areas,

and forest ages on the spatial variability of ecosystem resistance to heat and drought events.

## 2 Data and Methods

### 2.1 Satellite data

Table 1 presents an overview of the vegetation datasets included in this study. We used vegetation optical depth (VOD) data from ESA's SMOS low-frequency passive microwave sensor (SMOS L-VOD) as an indicator of vegetation dynamics

(Wigneron et al., 2021). Vegetation optical depth (VOD) parametrizes the attenuation of the microwave radiation when passing through the vegetation layer accounting for the effects of both the woody and leafy components of the vegetation canopy. VOD varies with both, the mass of water contained in the canopy and the canopy structure. Different VOD datasets have been interpreted to be dominantly sensitive to vegetation biomass at an annual scale, based on the assumption that relative water content (RWC) in vegetation is stable from year to year (Brandt et al., 2018). Therefore, VOD has been used

as a proxy for above-ground biomass (Brandt et al., 2018; Fan et al., 2019). Since it is independent of any vegetation index, VOD is robust for application in ecology and climate change studies. Compared to traditional vegetation indices (NDVI, EVI) and Ku-, X- or C-VOD, L-band VOD (L-VOD) is less sensitive to saturation effects at high biomass densities and originates from the entire canopy, not just the top layer as for higher frequency VODs (Fan et al., 2019).

**Table 1.** Overview of the satellite products included for investigating vegetation dynamics.

| Variable | Dataset | Metadata period | Sampling (time, space) | Reference |
|---|---|---|---|---|
| SMOS L-VOD | SMOS | 2010-2020 | Monthly, 0.25º | (Wigneron et al., 2021; Yang et al., 2023) |

| NDVI | MODIS | 2000-2020 | 16-day, 0.05º | (Didan, 2015) |
|------|-------|-----------|---------------|---------------|
| EVI  | MODIS | 2000-2020 | 16-day, 0.05º | (Didan, 2015) |

The multi-year L-VOD products were filtered strictly (Yang et al., 2023) in order to remove effects of radio frequency interference (RFI) in some regions of the northern hemisphere. The strictly filtered L-VOD data with good quality were then reconstructed using a curve-fitting method used for $CO_2$ measurements (Thoning et al., 1989). The specific process is as follows: First, the L-VOD data were fitted using both a harmonic function (reflecting seasonal cycle) and a polynomial function (reflecting long-term variability). Second, the residuals of the fitting were further transformed using a Fast Fourier Transform and then filtered using low pass filters to track the remaining seasonal oscillation and long-term variations. The reconstructed L-VOD data can be separated into two time series including the long-term variability only and the seasonal oscillation only. The first mainly reflects vegetation carbon dynamics, while the latter is more affected by the seasonality in the vegetation water content.

SMOS L-VOD data are more complex to interpret when the ground is frozen (e.g., ice, snow), hence, we removed observations where the MODIS snow or ice cover fraction in the specified pixel is larger than zero. We also filtered out observations where the MODIS vegetation cover fraction is less than 0.05 to focus on vegetated land pixels. L-VOD is sensitive to the moisture content of vegetation, which may be altered by water stress (Konings et al., 2021). Wigneron et al. (2020) proposed that the yearly average of the moisture content of vegetation is roughly constant between years at a 25 km scale. To limit the impact of variations in water content in estimates of vegetation resistance, we used the yearly maximum VOD as a proxy of the annual biomass that occurs mostly in the wet month, because relative vegetation water content during wet months is likely to be relatively stable over 2010–2021, so that the annual maximum L-VOD changes are closely related to vegetation biomass changes (Qin et al., 2021). This method resulted in 11 yearly values of L-VOD from 2010 to 2020.

Two optical vegetation indices, NDVI and EVI data, acquired from NASA's Moderate Resolution Imaging Spectroradiometer (MODIS) instrument aboard the Terra satellite, were used to compare with the SMOS L-VOD product. The MODIS-derived NDVI and EVI datasets are temporally and spatially consistent as they are obtained from a single platform and sensor and are regarded as state-of-art proxies for green vegetation cover. We further calculated kNDVI, a non-linear generalization of the NDVI following Eq. (1):

$$kNDVI = tanh(NDVI^2) , \tag{1}$$

kNDVI has been shown to perform better than NDVI with stronger correlations with flux tower estimates of gross primary productivity and satellite retrievals of sun-induced fluorescence. It has been evaluated to be more resistant to saturation and

noise (Camps-Valls et al., 2021). Compared to NDVI, EVI is proposed to decouple the canopy background signals and reduce the atmospheric influence. It presents higher resistance to saturation over dense forest areas (Zeng et al., 2022).

## 2.2 Climate drivers

The climate variables were acquired from the ERA5 reanalysis product at 0.25º spatial resolution for the 1979–2020 period (Hersbach et al., 2020) (https://cds.climate.copernicus.eu/cdsapp#!/home). To assess the ecosystem sensitivity to droughts and temperature, we used monthly averaged volumetric soil water in four soil layers at depths of 0–7, 7–28, 28–100, and 100–289 cm, and 2 m air temperatures. We calculated total soil moisture (SM) using the depth-weighted average of the volumetric soil water in the four layers.

Drought severity was quantified using the probability $P(x < SM)$ of the kernel density estimate (KDE) fitted using the distribution of monthly total SM anomalies for the 1979–2020 time period. We used KDE to fit the distribution (Flach et al., 2018). The total SM anomalies were then linearly detrended and deseasonalized. A given month t was defined as a drought month when $P(x < SM$ anomalies t$) < 0.1$ (monthly SM anomalies are less than the 10th percentiles of the KDE fit).

## 2.3 Land cover and land management data

Given the uncertainties in the land-cover mapping (Hartley et al., 2017; Li et al., 2018), we used three global land cover maps in our study presented in Table 2. These were resampled from their original resolution to 0.25º spatial resolution for the 2010–2020 period to match the spatial resolution of L-VOD.

**Table 2.** Overview of the land cover products included in this study.

| Land cover | Metadata period | Sampling | Reference |
| --- | --- | --- | --- |
| MCD12Q1 | 2001-2020 | Yearly, 500 m | (Friedl, 2019) |
| Land Cover CCI | 1992-2020 | Yearly, 300 m | (ESA, 2022) |
| LUH2 v2h | 850-2100 | Yearly, 0.25º | (Hurtt et al., 2020) |

We derived land cover based on the International Geosphere-Biosphere Program (IGBP) scheme from the MODIS land cover map. The ESA Land Cover CCI product and LUH2 v2h use different land cover classifications. ESA's Land Cover CCI product provided 37 classes based on the United Nations Land Cover Classification System (UN-LCCS; Di Gregorio and Jansen, 2005). These were then converted to 14 plant functional types (PFTs) using the lookup table in Poulter et al. (2015). To better account for the intrinsic bias and classification differences among the three products, we further aggregated

the original classes into four vegetation categories (forests, shrublands, grasslands and croplands) according to Table 3. We categorized land cover in bins of 25% fraction of each biome and, for each pixel, assigned the land cover information with the highest agreement across datasets ($\geq 2$ agree). To guarantee that our results are not biased by the land cover change (e.g., deforestation), we excluded those pixels showing changes in the 25% bins during 2010–2020.

**Table 3.** Overview of the merged input classes for the four vegetation categories used in this study.

| Vegetation categories | MCD12Q1 | Land Cover CCI | LUH2 v2h |
|---|---|---|---|
| Forests | Evergreen Needleleaf Forests, Evergreen Broadleaf Forests, Deciduous Needleleaf Forests, Deciduous Broadleaf Forests, Mixed Forests | Tree Broadleaf Evergreen, Tree Broadleaf Deciduous, Tree Needleleaf Evergreen, Tree Needleleaf Deciduous | Forested primary land, Potentially forested secondary land |
| Shrublands | Closed Shrublands, Open Shrublands, Woody Savannas, Savannas | Shrub Broadleaf Evergreen, Shrub Broadleaf Deciduous, Shrub Needleleaf Evergreen, Shrub Needleleaf Deciduous | |
| Grasslands | Grasslands | Natural Grass | Managed pasture, Rangeland |
| Croplands | Croplands, Cropland/Natural Vegetation Mosaics | Crops | C3 annual crops, C3 perennial crops, C4 annual crops, C4 perennial crops, C3 nitrogen-fixing crops |

The LUH2 v2h dataset also provides land management information, for example, forested primary land and potentially

forested secondary land (Figure 1). It also provides wood harvest area as a fraction of the total grid cell area. We converted this to the fraction of wood harvest from forests (described below as forest wood harvest intensity) by dividing the wood harvest area by the forest area fraction of the total grid cell area. We limit this analysis to pixels with > 50% forest cover. We also used the global map of irrigation areas around 2005 at 0.0833º spatial resolution from FAO (Siebert et al., 2013). The latter map provides areas equipped for irrigation as the percentage of total grid cell area ($f_e$) and the percentage of area

equipped for irrigation that was actually irrigated ($f_a$), based on national census surveys or irrigation sector studies. We then assumed that the irrigation equipment is totally located in cropland and calculated the percentage of cropland actually

irrigated $f_{ac}$ as $f_a*f_c/f_c$, where $f_c$ is cropland percentage of grid cell area (Figure 1). To explore the influence of tree characteristics on their resistance to droughts, we used a global forest age map estimated from forest inventories, biomass and climate data at 0.00833° spatial resolution (Besnard et al., 2021).


All above datasets were resampled to 0.25° to make them comparable to the relatively coarse L-VOD data. For the global forest age map, we calculate the average of the forest age in pixels at 0.25°.

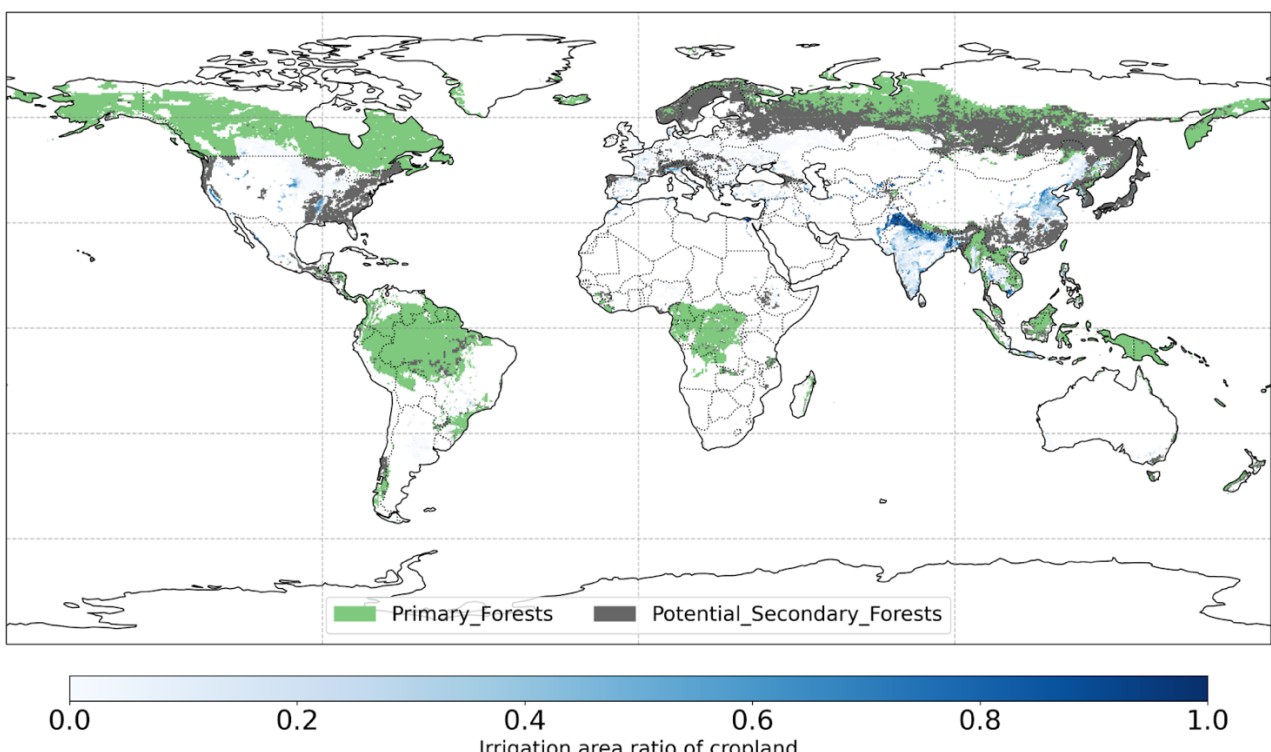

**Figure 1.** Global map of forest management types in LUH2v2 and irrigation area ratio of cropland for minimum MODIS cropland > 50% in 2011-2020.

## 2.4 Definition of ecosystem resistance

To calculate the ecosystem resistance to drought, we applied a linear model for each pixel following Eq. (2):

$$Y_{anom}(t) = \alpha N(t) + \beta T_{anom}(t) + c + \epsilon(t) , \tag{2}$$

for the year $t$ ranging from 2011 to 2020, in which $Y$ represents either yearly maximum L-VOD, yearly mean kNDVI or EVI. $N$ is the number of drought months in each year in 2010-2020. $T$ is the yearly mean 2 m air temperature. All anomalies were calculated through the subtraction of linear trends and through the subtraction of the average for each month over the

time period 2010–2020. c is the intercept term and $\epsilon$ is the residual term. The anomalies were then standardized as the following Eq. (3):

$$X' = \frac{X - \mu}{\sigma},$$     (3)

where $\mu$ is the average and $\sigma$ is the standard deviation of the anomalies.

We also analyzed the linear autoregressive model as Eq. (4):

$$Y_{anom}(t) = \alpha N(t) + \beta T_{anom}(t) + \varphi Y_{anom}(t-1) + c + \epsilon(t) ,$$     (4)

in which we also considered the memory effect with a lag-1 autoregressive (AR1) term, which has been demonstrated to be fundamental to understanding the daily carbon metabolism of terrestrial ecosystems (Liu et al., 2019a; Cranko Page et al., 2022) and ecosystem resilience of monthly NDVI (De Keersmaecker et al., 2016). We evaluated if interannual memory effects influence drought sensitivity, temperature sensitivity, and adjusted $R^2$, but the additional predictor does not carry new information for annual L-VOD, EVI, and kNDVI anomalies. The drought sensitivity and temperature sensitivity are similar

and adjusted $R^2$ does not increase in general (Appendix B), so we used the most parsimonious model Eq. (2).

We note that radiation plays an important role in the energy-limited boreal region. However, surface incoming solar radiation strongly correlates with temperature and drought duration. The air temperature at 2 m increases due to a higher energy input when surface incoming solar radiation is higher. Droughts are associated with clear-sky and sunnier conditions

that favor more incoming solar radiation (O et al., 2022). To avoid the influence of collinearity on estimated vegetation sensitivity to temperature and drought duration, and given that only ten years of data are available, we did not incorporate radiation into our linear regression model.

The model coefficient $\alpha$, also known as the sensitivity of $Y$ to $N$, can be closely related to the inverse of ecosystem

resistance during the drought period. We used $\alpha(month^{-1})$ as a metric for the ecosystem resistance to droughts. If $\alpha$ is negative, the vegetation L-VOD, as an indicator for above-ground vegetation biomass, and EVI, kNDVI as indicators for greenness are negatively disturbed during the drought period. On the contrary, if $\alpha$ is positive, the vegetation grows better during the drought period than in normal SM conditions. Higher $\alpha$ indicates stronger ecosystem resistance to droughts. Similarly, we used $\beta$ as a metric for the ecosystem sensitivity to temperature anomalies. This approach also partly controls

for the covariance between drought and temperature. $\alpha$ is in the unit of $month^{-1}$ but $\beta$ has no unit due to standardization. Therefore, we cannot directly compare $\alpha$ with $\beta$ to assess the effect of droughts and temperature anomalies. $\alpha$ and $\beta$ were evaluated for each pixel.

## 2.5 Statistical analysis

As described above, we characterized ecosystem resistance by the sensitivity of the vegetation state to the drought length and
2 m temperature anomalies. We further masked pixels where no droughts occurred in the 2010–2020 period. We aggregated
the values of ecosystem resistance for IPCC AR6 reference sub-regions (Iturbide et al., 2020), dominant IGBP vegetation
cover classes (Figure 3a), forest and cropland fraction, forest management types and irrigation percentages. We only
analyzed those pixels without changes in the dominant IGBP vegetation cover class during the period 2010-2020. We only
analyzed those pixels with less than 10% changes in primary and secondary forest, tropical evergreen broadleaf forests, and
crop cover fraction in 2010-2020. We used 10% to avoid abrupt and substantial changes in vegetation cover that might
directly modulate the above-ground vegetation biomass variations and therefore affect our regression. We compared the
distributions of these groups and distinguished the effect of increasing coverage of specific vegetation categories or some
specific land management.

As the secondary forests dominate mid latitudes and the primary forests dominate tropical and boreal regions, to minimize
the potential confounding environmental effects on the results, we extracted only pairs of primary and secondary forests
sharing similar long-term temperature and precipitation averages. First, we define primary forests as those pixels with > 50%
forest fraction and primary forests dominate > 50% of the forest fraction, similar for secondary forests where secondary
forests dominate > 50% of the forest fraction in 2010-2020. These pixels were then categorized according to ERA5
temperature and total precipitation long-term averages. Temperature is divided into 25 groups $T_i$ (i = 1–25) from -10 °C to
30 °C uniformly and precipitation is divided into 25 groups $P_j$ (j = 1–25) from 0 to 5000 mm uniformly, which results in 625
bins $T_iP_j$. Second, if there are more than 5 pixels of primary forests $PF_k$ and 5 pixels of secondary forests $SF_l$ in one bin $T_iP_j$,
the differences between the average of these pixels are calculated as $\Delta T_iP_j = \overline{PF_k} - \overline{SF_l}$. We then showed the distribution
of these differences for different bins $\Delta T_iP_j$ in the boxplot.

To compare the ecosystem resistance of forests with different mean ages, we selected pixels with dominant tropical
evergreen broadleaf forests, >50% forest fraction to avoid confounding effects of management over secondary forests. We
then selected only pixels belonging to the primary forests we defined above and grouped the forest ages into three groups [0,
100), [100, 300) and ≥ 300 years. We only selected pixels with over 50% forest fraction in 2011-2020 and no variation in the
dominant vegetation type (tropical evergreen broadleaf forests). To minimize the potential climate confounding effects on
the dependence of $\alpha$ on forest age, we limited our comparison in the tropics due to limited pixels with ≥ 300 years old trees
for other regions. We only compared pixels where crop fraction is > 50% in 2011-2020 for irrigation effects. We compared
the crop irrigation ratio between bins < 10%, 10% to 50% and ≥ 50% for pixels with > 50% cropland fraction, which is
dominated by cropland in India (Figure 1). For the comparison between two groups, we applied the unpaired two-sample
Wilcoxon test to test whether there is a significant difference between their medians (*P*-value < 0.05).

## 3 Results

### 3.1 Contrasting patterns of ecosystem resistance to droughts and temperature over different regions and dominant vegetation types

At the global scale, $\alpha$ based on Eq. (2) are negative in 55% of pixels with valid values from L-VOD, 56% from EVI, and 59% from kNDVI (Figure A6a-A6c, Figure 3a). Negative resistance values predominate in mid latitudes, while positive values are found for the boreal regions (Figure 2d). In the tropics, we find the largest divergences between the three indices, where drought resistance based on L-VOD is positive in Amazon and Southeast Asia regions, but in Southeast Asia, EVI shows the lowest drought resistance with a median of -0.12 $month^{-1}$ in disagreement with L-VOD which has a median of 0.03 $month^{-1}$. kNDVI shows the same sign as EVI, with a median of -0.05 $month^{-1}$. In the Amazon, there are around 15% and 18% more pixels showing negative resistance from EVI and kNDVI than L-VOD. In the Amazon region, there are around 15% and 18% more pixels showing negative resistance from EVI and kNDVI than L-VOD. In our analysis, we observe that 12% of pixels show significant drought resistance at a 10% significance level (6%, 6%, and 7% at a 5% significance level) from L-VOD, EVI, and kNDVI. We only used the significant drought resistance at a 5% significance level to investigate the impacts of land cover and land management, ensuring that the vegetation growth is impacted by drought conditions. The standard error of the drought resistance coefficient is relatively higher for tropical regions with higher uncertainties in the edge of the Amazon and central Africa forests (Figure A6e-A6h).

The temperature sensitivity shows a clearer spatial pattern than the resistance to drought duration. The ecosystem resistance to temperature is negative and lower in tropical regions compared to mid latitudes (Figure 2l), indicating a decline in above-ground vegetation biomass and greenness during hot weather and a weaker ecosystem resistance to high temperatures in tropical regions. In boreal regions, the ecosystem resistance is mostly positive, which indicates vegetation growth during hot weather and thus a stronger resistance to hot events. In the tropics, the forest regions and non-forest regions show divergent resistance, with forest regions showing mostly positive resistance to temperature, and negative values predominating in grassland or savannas. L-VOD shows a large deviation relative to kNDVI and EVI in boreal regions, mostly because of missing data for L-VOD in those regions. 15%, 26%, and 31% of pixels show significant temperature sensitivity at a 10% significance level (9%, 17%, and 21% at a 5% significance level) from L-VOD, EVI, and kNDVI. We only used these pixels to investigate the land cover and land management effects to make sure that the vegetation growth is relevant to temperature. The standard error is also relatively higher for the Northern Hemisphere with higher land cover fraction but more pixels with missing values (Figure A6m-A6p).

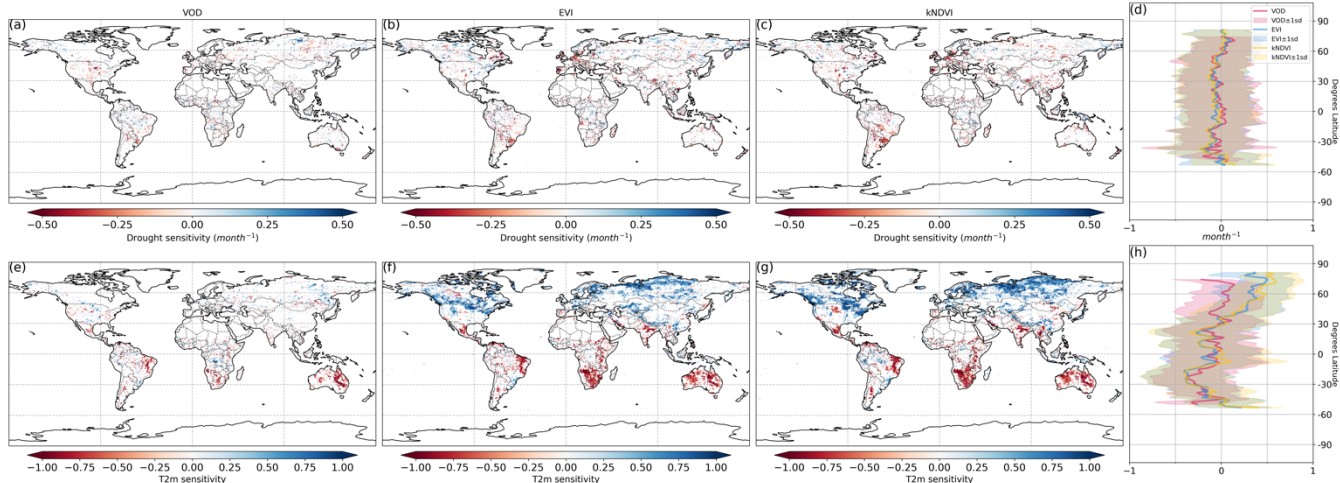

**Figure 2.** Ecosystem resistance to drought duration and temperature sensitivity. Spatial map of drought resistance $\alpha$ for (a) L-VOD, (b) EVI, (c) kNDVI. Same for temperature sensitivity $\beta$ (e, f, g). The averages for different latitudes and their standard deviations are shown on the right (d, h). The pixels with non-significant $\alpha$ and $\beta$ at a 10% significance level are masked with white color. The full-page figures where pixels with non-significant $\alpha$ and $\beta$ at a 5% significance level are masked with white color are provided in the supplementary Figure S1-S3 for better visualization.

Our model performs better in some areas in eastern South America, southern Africa, eastern Australia, and some boreal regions, where the $R^2$ is higher than 0.5 (Figure A7). At the global scale, approximately 15%, 23%, and 27% of the pixels exhibit an $R^2$ exceeding 0.5 when derived from L-VOD, EVI, and kNDVI, respectively.

We summarize the results for each of the IPCC AR6 land sub-regions in Figure 3. The full name of each land sub-region is shown in Table A1. 32 regions coloured red or blue in Figure 3a have mean values of $\alpha$ significantly different from zero, as determined by the two-tailed Student t-test (*P*-value < 0.05). The median values of $\alpha$ range from -0.20 to 0.07 $month^{-1}$ across regions, and the average values from -0.25 to 0.08 $month^{-1}$. Among these regions, 25 regions show significant negative mean $\alpha$. Only NWN, NEU, RAR, and RFE in the boreal region, NSA, SAM and SEA with high forest cover fraction show positive mean $\alpha$. By contrast, most regions over mid-latitudes (30°–60°) show negative mean $\alpha$ (Figure 3a, 3b).

In the tropics, SAS, CAF, WAF, and NAU, which have lower forest cover show negative mean $\alpha$, while NSA, SEA and SAM, which are dominated by evergreen broadleaf forests, show higher resistance and positive mean $\alpha$. We grouped $\alpha$ by different IGBP vegetated land cover types from forests to cropland. $\alpha$ based on L-VOD are higher in evergreen needleleaf forests (ENF), shrublands (SH) and evergreen broadleaf forests (EBF), but lower in cropland (C), deciduous needleleaf forests (DNF), and crop/natural vegetation mosaic (CNVM) (Figure 3c). kNDVI and EVI agree on the lower resistance in C and CNVM and high resistance in SH, ENF, and mixed forests (MF) (Figure A3).

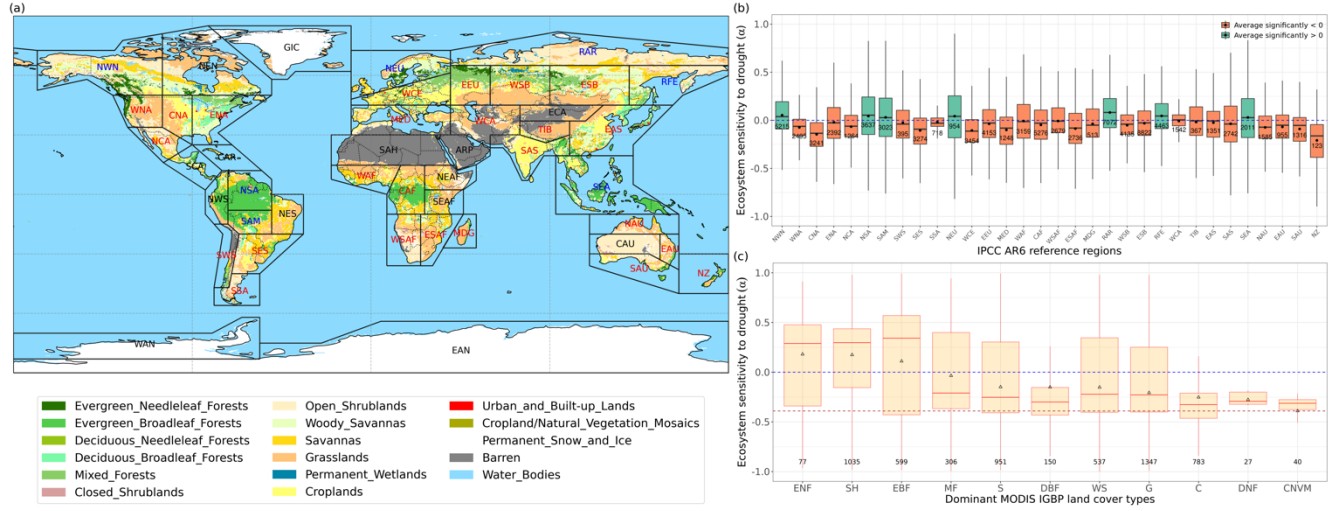

**Figure 3.** Regional pattern of ecosystem resistance to drought duration. (a) Distribution of the IGBP land cover types based on MODIS (MCD12Q1). The boxes with abbreviations indicate updated reference regions for IPCC AR6 WG1. Blue represents significant positive mean of $\alpha$ and red represents significant negative mean of resistance, black indicates regions whose means of the distribution are not significantly different from zero ($P$-value < 0.05); (b) Distribution of ecosystem resistance $\alpha$ to drought for different reference sub-regions,
the number in each box is the number of pixels in this category. Only regions with averages that are statistically different from zero are shown (two-sided Student $t$-test; $P$-value < 0.05); (c) Distribution of ecosystem resistance to drought $\alpha$ for different dominant IGBP vegetation classes. Only significant $\alpha$ from the linear model is selected ($P$-value < 0.05) are selected.

## 3.2 The importance of forest cover fraction in modulating ecosystem resistance

A given pixel can have a mixture of several land cover types with very similar fractions, but the dominant land cover pattern ignores this effect. Therefore, we further analysed mean $\alpha$ among different forest and crop cover fractions. In general, the mean $\alpha$ is lower (more negative) with decreasing forests and increasing cropland fraction. Pixels dominated by forests are significantly ($P$-value < 0.05; indicated by stars) more resistant to droughts than those where cropland predominates (Figure 4a-c). $\alpha$ increases with increased forest fraction, from a mean value of -0.07 $month^{-1}$ (0-25% forest cover) to 0.07 $month^{-1}$

(75-100% forest cover), and decreases with increased cropland fraction, from -0.07 $month^{-1}$ for 0-25% crop cover to -0.30 $month^{-1}$ for 75-100% crop cover. The resistance is close to zero because pixels with less than 25% forest and less than 25% cropland are dominated by low-vegetated land or bare soil, so that the signal of vegetation is weak.

In forest-dominated regions (> 50% forest fraction), the difference between results for L-VOD, EVI and kNDVI becomes
more obvious than in crop-dominated regions. Ecosystem resistance is the highest and even positive for L-VOD and the lowest for EVI, and kNDVI shows similar ecosystem resistance to that of L-VOD.

The contrast between forest-dominated regions and crop-dominated regions also exists for ecosystem resistance to 2 m air temperature from EVI and kNDVI (Figure 4e-f). The significant positive sensitivity to temperature predominates in the

regions with more than 25% forests while the sensitivity is negative in the regions with more than 50% cropland. The pattern is distinct for different climate zones and shows strong latitudinal dependence. For tropical regions, except for kNDVI in a small fraction of regions, most ecosystems show negative resistance to temperature, which means that higher temperatures lead to a negative impact over a large area in the tropics (Figure A5a-c). In temperate climate zones, predominant in mid latitudes, all three vegetation products agree on the negative resistance in the regions with less than 25% forest cover, but kNDVI and EVI show a higher positive resistance in the regions with more than 25% forest cover (Figure A5d-f). In the boreal region, the resistance values are generally positive (Figure A5g-i), which confirms the variability of resistance to temperature to latitudes.

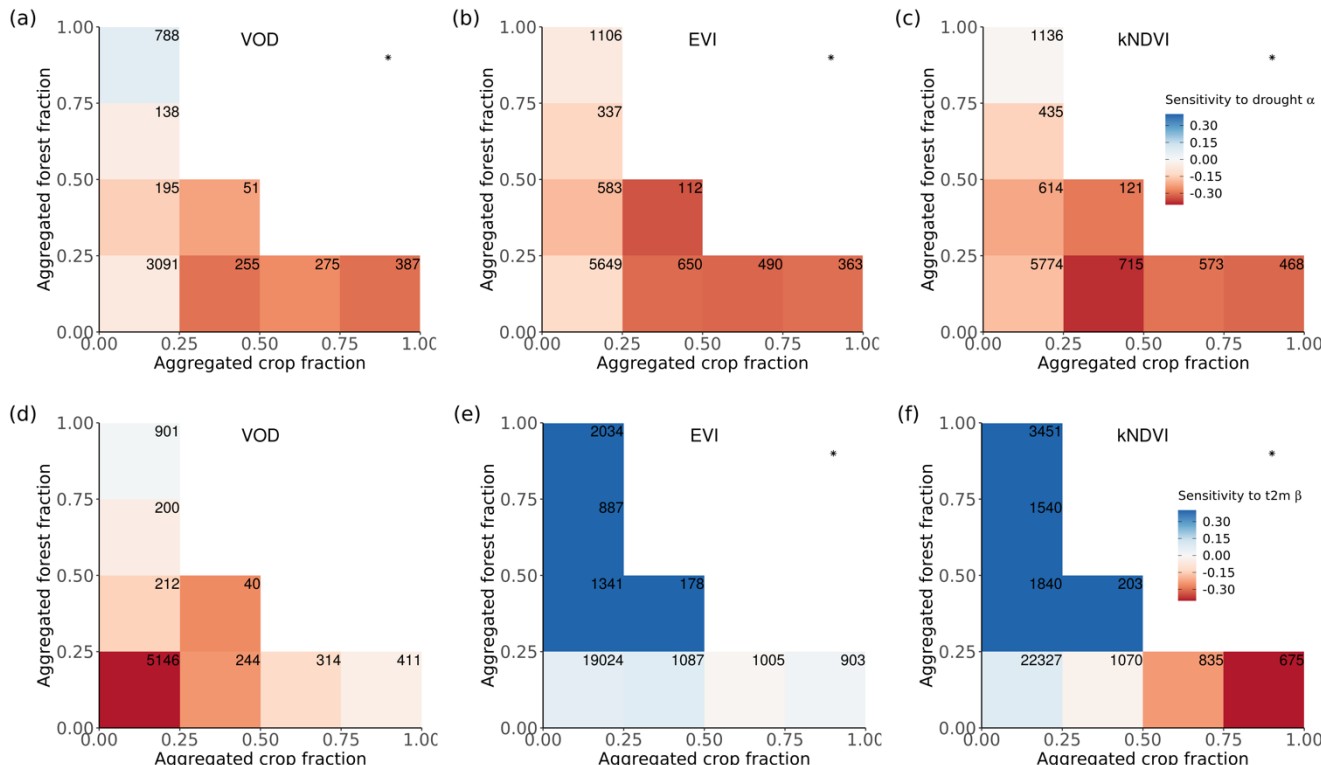

**Figure 4.** Ecosystem resistance to drought and temperature binned for different levels of the aggregated forest and cropland fraction classes from the three land cover products (a) L-VOD, (b) EVI and (c) kNDVI for drought resistance coefficients $\alpha$ and (d-f) for temperature resistance coefficients $\beta$. Only significant coefficients $\alpha$ in the linear model (P-value < 0.05) are included and groups with less than 20 pixels are excluded. The number in each bin is the number of pixels in this category. Only pixels with no change in 25% bins of the four dominant vegetation categories (forests, shrublands, grasslands and croplands) are analyzed. Star on the upper right corner indicate significantly higher resistance in forest > 75% than crop > 75% at the 0.05 significance level from the unpaired two-sample Wilcoxon test.

### 3.3 The roles of forest management, crop irrigation and forest ages in modulating ecosystem resistance

The above results suggest that a transition between dominant vegetation types modifies ecosystem resistance. Other anthropogenic disturbances, such as forest management, have the potential to influence the ability of forest ecosystems to maintain their functioning during drought-heat extremes by directly affecting tree species, age distribution, cover density, rooting depth and primary productivity. For croplands, irrigation is also an essential element to maintain and increase yields during droughts.

To detect such effects, we analysed ecosystem resistance to droughts $\alpha$ for primary and potential secondary forests from the LUH2 v2h land cover dataset (Figure 5a). It should be noted that to minimize the potential effects of the fact that primary and secondary forests are distributed differently across climate zones, we compare pairs of pixels with primary and secondary forests under similar temperature and precipitation climatological conditions bins as described in section 2.5. The comparison between primary and potential secondary forests shows that the averaged ecosystem resistance $\alpha$ calculated from L-VOD is significantly higher in primary forests than in secondary forests. The median of the difference between primary and secondary forests ($\Delta\alpha$) is 0.382 $month^{-1}$ and is significantly greater than 0 based on the one-sample Wilcoxon test ($P$-value < 0.05). However, we did not detect such a large difference between EVI and kNDVI, whose medians of $\Delta\alpha$ between forest types are -0.040 $month^{-1}$ and 0.052 $month^{-1}$, but given the large spread of the distribution, their medians are not significantly greater than 0 based on the one-sample Wilcoxon test ($P$-value > 0.05).

We further tested the effect of forest ages in modulating the ecosystem resistance in the tropical primary evergreen broadleaf forest. Based on L-VOD, forests older than 100 years are substantially more resistant to drought than forests younger than 100 years. The median of $\alpha$ for forests younger than 100 years is -0.549 $month-1$, while the median values of $\alpha$ for forests aged 100-300 years and older than 300 years are 0.455 $month-1$ and 0.360 $month-1$ respectively. We also find a significant ($P$-value < 0.05) increase of $\alpha$ in kNDVI between forests aged 100-300 years and older than 300 years, but the effect is not as large as in L-VOD and we found no significant differences based on EVI. These results indicate that VOD is more sensitive to water volume and biomass than reflectance indices in general.

We also investigated the ecosystem resistance $\alpha$ for different irrigation levels (Figure 5c). The result shows also an increasing resistance for L-VOD with the irrigation levels, with the median of $\alpha$ for L-VOD increasing from -0.342 $month^{-1}$ to 0.023 $month^{-1}$ between less than 10% actually irrigated cropland and more than 50% actually irrigated cropland significantly ($P$-value < 0.05). For kNDVI and EVI, the change in the median of $\alpha$ is negligible but we still found a higher percentage of pixels with close to zero or positive resistance for higher irrigation fractions.

We finally explored the potential role of forest wood harvest intensity (Figure 5d). All three satellite products agree on a
significant decrease of drought resistance ($\alpha$) with increased forest wood harvest intensity, from a median of -0.21 $month^{-1}$
under < 1% harvest area ratio, to -0.34 $month^{-1}$ under 1-10% wood harvest intensity, and -0.40 $month^{-1}$ for >10% harvest
intensity, based on L-VOD. Results from EVI and kNDVI are consistent with those of L-VOD.

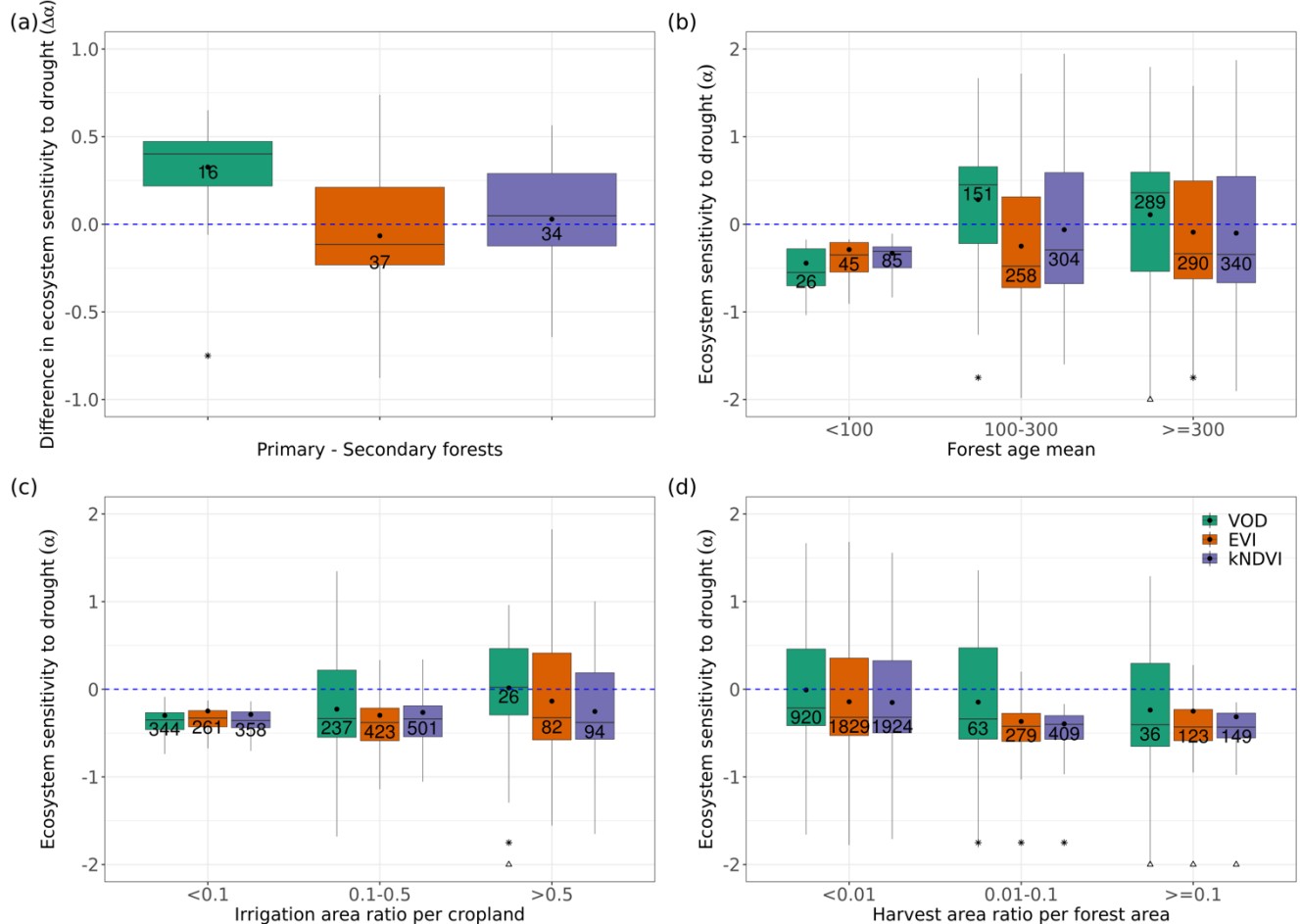

**Figure 5.** Ecosystem resistance to droughts for (a) forest management located in a similar background climate, and for different levels of
(b) forest ages in the tropical primary EBF, (c) crop irrigation and (d) forest harvest area fraction. Only significant coefficients $\alpha$ in the
linear model (*P*-value < 0.05) are included. Stars in (a) indicate the median value of this category is greater than 0 at the 0.05 significance
level from the one-sample Wilcoxon test. Stars indicate the median value of this category is greater than the median of the previous
category and triangles indicate the median of this category is greater than the median of the first category at the 0.05 significance level
from the unpaired two-sample Wilcoxon test. The number in each box is the number of bins/pixels in this category. Only pixels with
unchanged dominant primary and secondary forest, tropical primary EBF, and crop cover in 2011-2020 were selected, as defined in
Section 2.5.

# 4 Discussions

## 4.1 Spatial variability of ecosystem resistance to drought and temperature

Ecosystem resistance during a disturbance phase plays a key role in determining resilience. In this study, we evaluated the ecosystem resistance to drought and temperature with their sensitivity to 10th percentile soil moisture drought duration and temperature anomalies at 2 m in a linear model. The IPCC AR6 sub-regions, whose averages of drought resistance are significantly negative, mainly correspond to semi-arid regions with high coverage of grassland, cropland and savannas and temperate deciduous forests (Figure 3a). In these semi-arid regions dominated by grassland, cropland and savannas, vegetation dynamics is more sensitive to water availability and therefore is more impacted by drought events (Poulter et al., 2014; Ahlström et al., 2015; Walther et al., 2019). In temperate forests over mid latitudes, previous studies have suggested that short seasonal droughts are more likely to induce dieback of broadleaved deciduous angiosperm trees (e.g., DBF in Figure 3c) than conifer trees (i.e., ENF in Figure 3c) because of the higher vulnerability to xylem cavitation of angiosperm trees (Maherali et al., 2004; Allen et al., 2010). On the contrary, areas with significantly positive resistance to drought correspond to dense tropical evergreen broadleaved forests and boreal continental regions with shrublands and evergreen needleleaved forests. For boreal regions, the soil is generally humid, so that the drought defined as the 10th percentile likely still provides critical water storage for vegetation. The potential environmental limitations to vegetation growth in these areas are temperature and radiation rather than water availability (Boisvenue and Running, 2006).

We also evaluated whether limiting drought duration to the growing season of each year. The resulting $\alpha$ and $\beta$ values over pixels where the coefficients are significant ($P$-value < 0.05) are strongly correlated to $\alpha$ and $\beta$ calculated based on whole-year drought duration and results still hold.

The resistance to temperature highly depends on latitude. For extratropical regions, vegetation L-VOD and greenness normally increase with higher temperatures. Warming generally leads to an earlier growing season onset and results in increased early-season vegetation productivity (Forkel et al., 2016). Although we removed the confounding effect of extreme soil water deficit from L-VOD, this trend can still be adversely affected by climate variability and other emerging limitations from energy and nutrients on vegetation production (Piao et al., 2017; Buermann et al., 2018; Liu et al., 2019b).

## 4.2 Contrasting ecosystem resistance to drought and temperature in forests and cropland

Forests and cropland respond differently to extreme drought events. Ecosystem resistance increases with increased forest and decreased cropland fraction. This is also observed in previous studies based on GPP datasets over Europe (Zhang et al., 2016; Bastos et al., 2020) and SIF (Walther et al., 2019). The increasing pattern in L-VOD for dominant forests might be a result of increased insolation and photosynthetic activity light, and weak changes in greenness during the drought period (Zhu et al., 2018; Walther et al., 2019). Besides, the light use efficiency (LUE) decreased less with lower soil moisture

contents in forests compared to non-forest vegetation, which could be linked to deeper and more extensive root systems with higher access to available soil water (Walther et al., 2019). The pattern is consistent for the tropical, temperate and continental climate regions except for results from EVI in the temperate climate (Figure A4) and can be explained by the intrinsic structural and physiological differences between trees and crops, for example, the deeper rooting depth of trees (Canadell et al., 1996), higher water storage capacity in the stems for forests (Matheny et al., 2015) and different water use strategies between forest and grass-/cropland (Teuling et al., 2010).

Compared to drought resistance, the ecosystem resistance to temperature shows a weaker contrast in L-VOD between forests and cropland, instead, latitude dependence plays a more important role here (Figure A5). Nevertheless, from EVI and kNDVI, we still observe a divergent response to temperatures between forest-dominated regions and crop-dominated regions in tropical and temperate climate zones. In the temperate regions, forests may benefit from higher temperatures through warming-induced changes in their phenology, while crops might show a nonlinear response of photosynthesis to temperature, due to a weaker resistance to hot extreme days. A strong trend of earlier spring growing season onset and later autumn senescence has been observed in the temperate forests in the eastern USA (Keenan et al., 2014). In boreal climate zones in the Northern Hemisphere, L-VOD, EVI and kNDVI generally show a strong positive relationship with an increase in temperature, which can be interpreted as an increase in photosynthesis in response to warming when enough water is available (Piao et al., 2006).

At the same time, croplands can be affected by different management practices, for example, crop rotation that changes from year to year and the variable timing of planting and harvesting also has an influence on vegetation biomass. By taking the yearly maximum L-VOD value, we partly alleviate such an influence and expect interannual variations to better correspond to biomass changes.

## 4.3 Effects of land management and forest age on ecosystem resistance

After accounting for the potential effect of climate background, primary forests still show significantly higher resistance than the potential secondary forests from L-VOD (Figure 5a). The pixel number is different due to a different ratio of significant resistance for primary and secondary forests located in similar climate backgrounds. Primary forests have substantially higher biodiversity values compared to secondary degraded forests even after partly accounting for confounding colonization and succession effects from the isolation, composition of surrounding habitats, and time since disturbance (Gibson et al., 2011). High tree species diversity helps strengthen the ecosystem resistance to droughts (Liu et al., 2022). Primary forests also show higher hydraulic diversity than secondary forests, which buffers impacts in ecosystem flux during dry periods across temperate and boreal forests (Anderegg et al., 2018). Secondary forests in the Brazilian Amazon are vulnerable to drought stress with a lower carbon balance and growth rates, and they only reached 56% of the tree diversity in the nearest primary forests (Elias et al., 2020). In Amazon forests, forest greening in degraded forests disturbed by fire has been found to

be more dependent on water resources than in mature forests (Roux et al., 2022). The new forest edges in much more fragmented degraded forest landscapes increase canopy desiccation, tree mortality, and fire frequency (Briant et al., 2010; Broadbent et al., 2008), especially during drought events (Roux et al., 2022). In boreal forest ecosystems in Sweden, primary forests have been found to be less affected by drought compared to secondary forests (Wolf et al., 2023). Primary forests also likely harbor older trees, which also show higher resistance to drought (Figure 5b). Besides, primary forests might have a more extensive rooting system with higher availability of soil water. However, it remains difficult to disentangle the above factors for the complex ecosystem due to limited data.

Apart from the effect from land cover due to different sensitivity to drought stress for different vegetation types, at the ecosystem scale, we illustrate that over tropical EBF, older trees tend to be more resistant to droughts (Figure 5b). Young trees sometimes exhibit high drought-induced mortality rates due to limited rooting depth (McDowell and Allen, 2015). Young fast-growing and light-wooded trees are recorded to be especially vulnerable to drought by cavitation or carbon starvation (McDowell et al., 2008; Phillips et al., 2009). Older mature forests could develop a more complex ecosystem with higher species diversity (e.g., Amazon rainforest). Tree species diversity may enhance the drought resistance in global forests with a stronger effect over tropical forests (Liu et al., 2022).

Irrigation helps attenuate drought impacts and enhance the cropland resistance to drought extremes. This effect is also reflected in SIF and GPP anomalies (Gampe et al., 2021; Cheng et al., 2022). It increases the mean SM during drought events and alleviates drought-heat stress through an increase in total evapotranspiration because it will increase atmospheric water vapour amount and decrease the mean surface daytime temperature (Mueller et al., 2016).

Our results indicate that forests with higher harvest intensities tend to be less resistant to drought globally. In-situ studies in different biomes show that forest management can influence forest resistance to disturbances such as drought (Silva Junior et al., 2020; Fawcett et al., 2022). This could be linked to the more complex structure of dense forests, whose below canopy microclimate might help to buffer forest stands from macroclimatic temperature extremes, e.g., in temperate broadleaved and mixed forest biome (Sanczuk et al., 2023). Forest thinning, depending on its intensity, has also been reported to result in lower drought resistance and resilience in older mature forests in north temperate forest ecosystems. This might be due to trees reaching larger sizes during stand development, which in turn increases water demand during droughts (D'Amato et al., 2013).

## 4.4 Ecosystem resistance difference between kNDVI, EVI and L-VOD

L-VOD responds differently to drought stress with EVI and kNDVI, especially in dense forests. As shown in Figure 4, the mean of ecosystem resistance is closer for dominant cropland but differs more for the dominant forests for the three products. Similarly, enhanced ecosystem resistance is not detected in kNDVI and EVI for primary forest (Figure 5a), which

is supposed to be denser with a more complex canopy structure than secondary forest. Such discrepancies can be related to the intrinsic difference in measurements between L-VOD and traditional vegetation indices. ESA's SMOS L-VOD product is caculated from low-frequency, large-wavelength microwave emissions. It has superior sensitivity to carbon density than NDVI, EVI and other higher-frequency VOD products. L-VOD signals originate from deeper volumes of a multi-layer canopy and thus correspond better to the above-ground biomass (Rodríguez-Fernández et al., 2018; Tian et al., 2018; Fan et

al., 2019; Wigneron et al., 2020). As a result, it is better capable to retrieve the overall biomass in dense tropical ecosystems, when EVI, NDVI and high-frequency L-VOD saturate (Liu et al., 2015). kNDVI overcomes the greenness saturation with increased forest cover empirically, but still does not detect biomass change under a top canopy layer other than leaf biomass. Generally in dense forests, L-VOD is sensitive to woody biomass where EVI and kNDVI only detect op-layer canopy greenness dynamics. Therefore, correlation between L-VOD and kNDVI is much lower in forests than in cropland and

grassland (Figure 6). This is confirmed in this study in the difference between L-VOD, EVI and kNDVI in dense forests under the tropical climate (Figure A4a-c), where canopy structure is generally more complex and taller. However, the ecosystem resistance to droughts is similar for dominant forests under continental climate (Figure A4g-i), because the forest canopy structure is generally simple and the forest canopy height is lower.

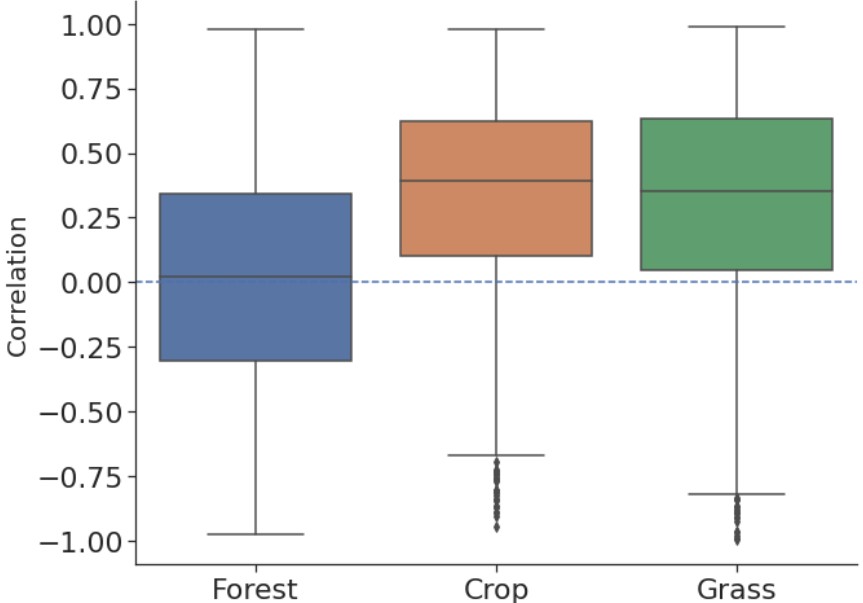

**Figure 6.** Correlation between yearly maximum L-VOD and kNDVI for different land cover types.

### 4.5 Limitations and outlook

Our method assumes a close relationship between L-VOD and vegetation biomass at an annual scale, as other studies using VOD products (Tian et al., 2017; Brandt et al., 2018; Fan et al., 2019; Wigneron et al., 2020; Qin et al., 2021). This

assumption is questioned by Konings et al. (2021) and they showed a weak temporal correlation between biomass anomalies calculated from a new biomass dataset from Xu et al. (2021) and L-VOD anomalies. Yet, the ability of the new dataset to represent the inter-annual variability of biomass has not been tested. Several studies also reported a strong near-linear relationship between L-VOD and biomass for woody vegetation independent of the year (Rodríguez-Fernández et al., 2018; Brandt et al., 2018), indicating a relatively constant RWC. In our study, we used the annual maximum L-VOD values in

order to minimize potential confounding effects by variations in vegetation water content. The comparison between primary and secondary forests from yearly maximum L-VOD fits better with in situ studies than EVI and kNDVI (Nunes et al., 2022), which reveals the potential of L-VOD to detect vegetation dynamics, especially in dense forests. To better disentangle the effect of RWC on L-VOD, continuous biomass measurements are required.

Even though we controlled for similar climate backgrounds by aggregating pixels based on their long-term temperature and precipitation averages, there are other climate effects that were not considered in our statistical analysis, for example, the interannual variability of precipitation and climate seasonality of temperature. With only limited areas exhibiting significant drought resistance $\alpha$, and given the need to ensure a large enough number of pixels for comparison in a similar climate space, it remains challenging to disentangle the potential confounding effects of all the climate variables and their

variabilities.

Drought duration shows a high correlation with yearly mean temperature in some regions in northern Amazon, Southern Africa and South Asia (Figure A8), so the multiple linear regression model might not perfectly disentangle their effects in these areas. We avoid these issues by analyzing those pixels with significant values of $\alpha$ and $\beta$ ($P$-value < 0.05).


Other factors related to land management, e.g., different crop rotations or harvest intensities, also play an important role in changing the vegetation biomass or greenness, especially in croplands, and might influence drought resistance and temperature sensitivity. The LUH2 v2h dataset provides additional information about crop and wood harvest practices. Crop harvest in LUH2v2 is spatially homogeneous so that it cannot be used to evaluate spatial differences in drought and

temperature sensitivity over croplands. Forest wood harvest in LUH2 v2h is smaller than 1% of the respective forest area for more than 90% of vegetated pixels (vegetation cover $\geq$ 5%). Therefore, we tested that the effect on our main results for primary forests and forest age is residual. For a more detailed analysis of other management practices, higher-resolution data on vegetation and management would be needed.

In general, regions showing significant negative mean values of drought sensitivity $\alpha$, which indicate negative impacts on vegetation during drought period, are mostly located in water-limited regimes (Denissen et al., 2022) except for EAS with relatively small coverage of available L-VOD data and CAF which is less homogeneous. Under climate change, the widespread shift from an energy-limited regime to a water-limited regime (Denissen et al., 2022) will put more ecosystems

under threat of droughts. The effects of specific land cover and land management on drought resistance are thus important for these regions in the future. Deforestation and a transition to cropland might potentially weaken the ecosystem resistance to extreme drought events. The protection and maintenance of primary forests are also important to sustain tropical biodiversity and its high drought resistance. For cropland, a higher irrigation ratio significantly increases the ecosystem resistance, but further precise and efficient practices of irrigation for agriculture are required to avoid a waste of water resources (Shulka et al., 2019).

## 5 Conclusion

This study analyses how land cover and common land management practices modulate the ecosystem responses to drought and heat based on different remote sensing products, namely L-VOD, EVI and kNDVI. Areas with predominant forest cover show stronger ecosystem resistance to extreme soil droughts than those predominated by croplands. Forests do not show obvious changes in canopy greenness indices during dry conditions compared to normal conditions, while L-VOD, as a proxy for biomass, tends to show a slight increase. This is possibly because of enhanced photosynthesis activity and a smaller decline in LUE compared to crops. Distinct responses are found between primary forests and secondary forests from L-VOD. Primary forests, typically associated with higher biodiversity, tend to show stronger resistance to droughts than secondary forests. Our findings from L-VOD show that tree age potentially contributes to the difference in drought resistance in tropical EBF, with ecosystems with older trees better mitigating drought stress.

Our results show the advantage of L-VOD in detecting vegetation dynamics in dense forests where EVI and kNDVI only detect the upper leaf canopy, which can therefore be a promising approach. In summary, we found that canopy greenness correlates well with canopy biomass and photosynthesis for nonwoody vegetation, forest biomass may fluctuate when canopy greenness is relatively stable. Irrigation helps improve the ecosystem resistance for croplands. Deforestation and afforestation leading to a change in forest cover and primary forest destruction might therefore modulate regional changes in ecosystem resistance. Forest management changes forest age distribution, possibly modulating the response of forests to droughts. The effect of forest management and crop irrigation on ecosystem resistance has important implications for the monitoring and management of ecosystems under climate change.

 **Appendix A: Supplementary Figures and Tables**

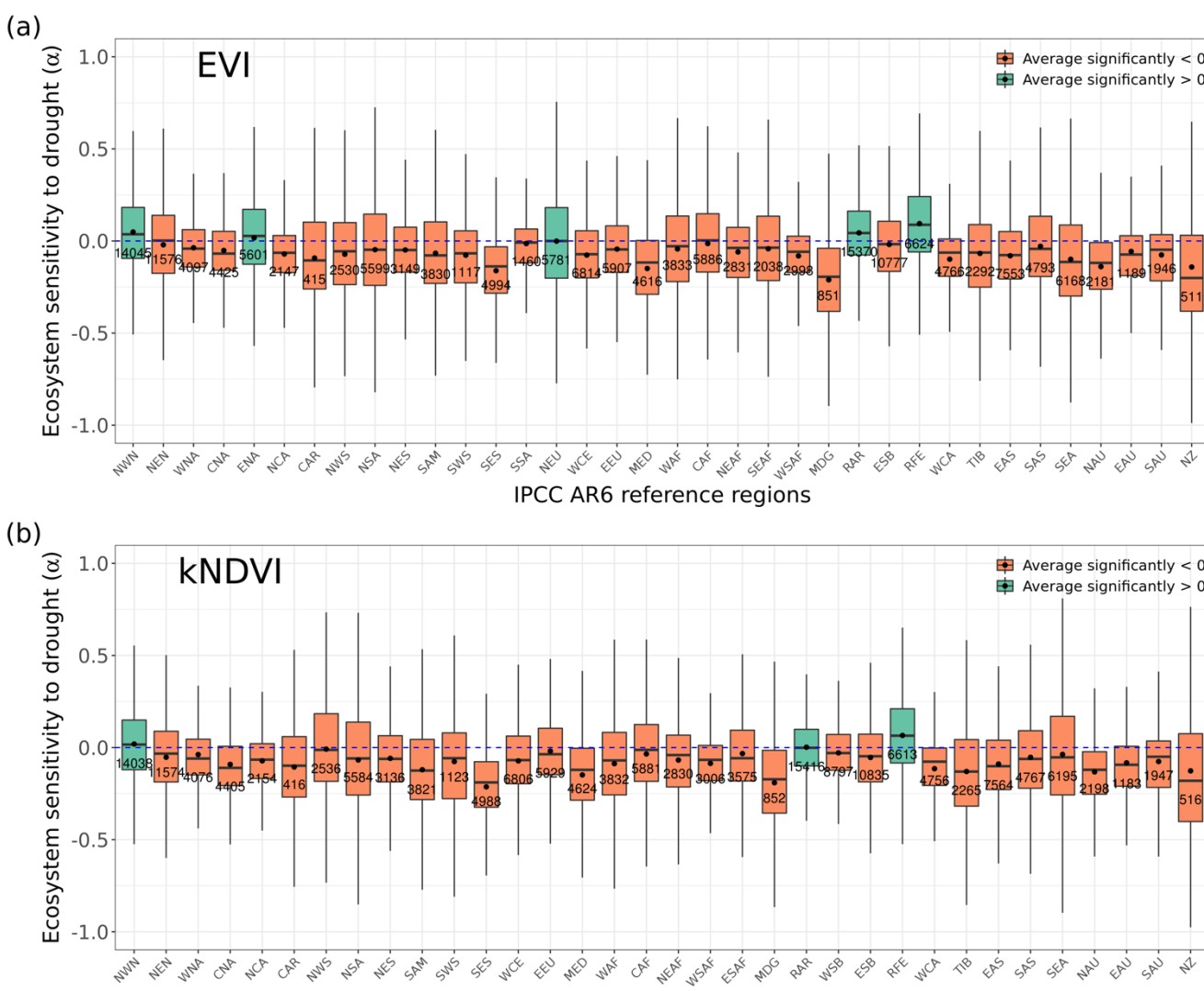

**Figure A1.** Distribution of ecosystem resistance $\alpha$ from (a) EVI and (b) kNDVI to drought for different dominant IPCC AR6 sub-regions. The boxes with abbreviations indicate updated reference regions for IPCC AR6 WG1. Blue represents significant positive resistance and red represents significant negative resistance, the number in each box is the number of pixels in this category.

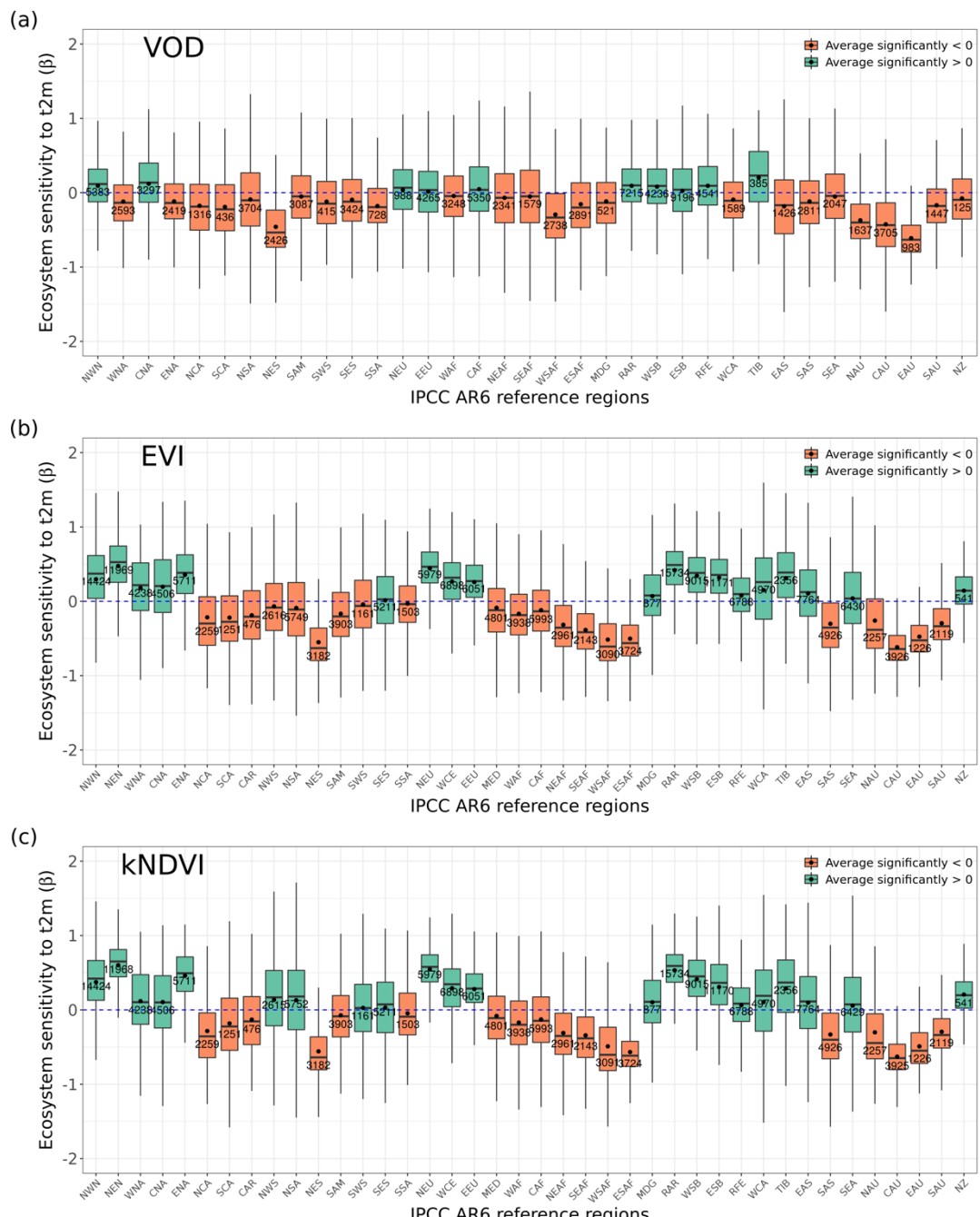

**Figure A2.** Distribution of ecosystem sensitivity to 2 m air temperature $\beta$ from (a) L-VOD, (b) EVI and (b) kNDVI to drought for different dominant IPCC AR6 sub-regions. The boxes with abbreviations indicate updated reference regions for IPCC AR6 WG1. Blue represents significant positive sensitivity and red represents significant negative sensitivity ($P$-value < 0.05), the number in each box is the number of pixels in this category.

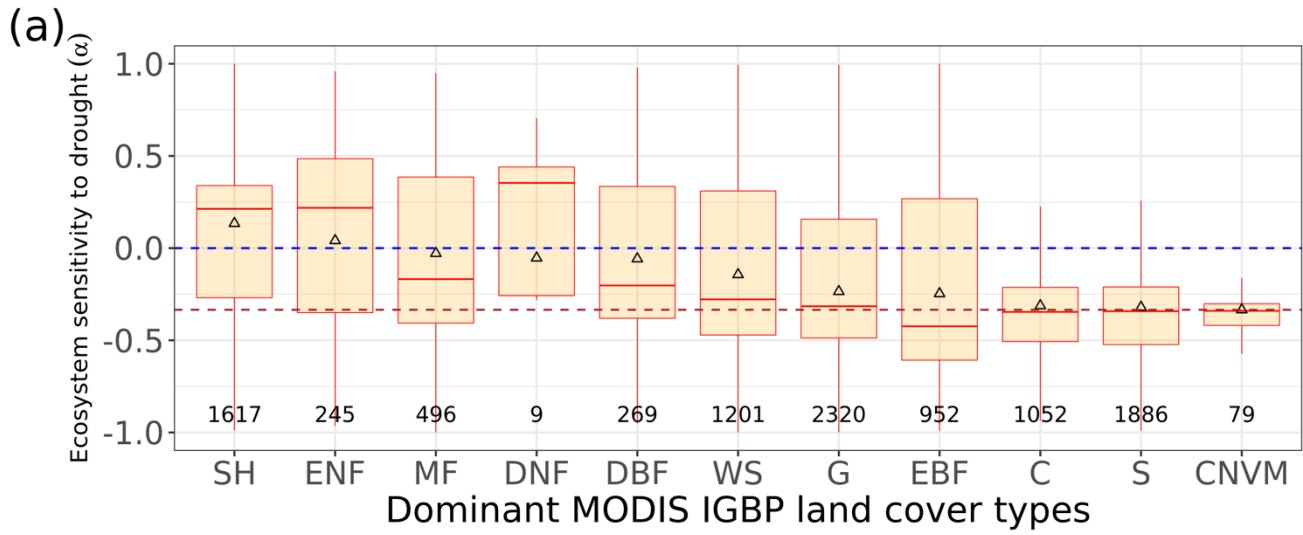

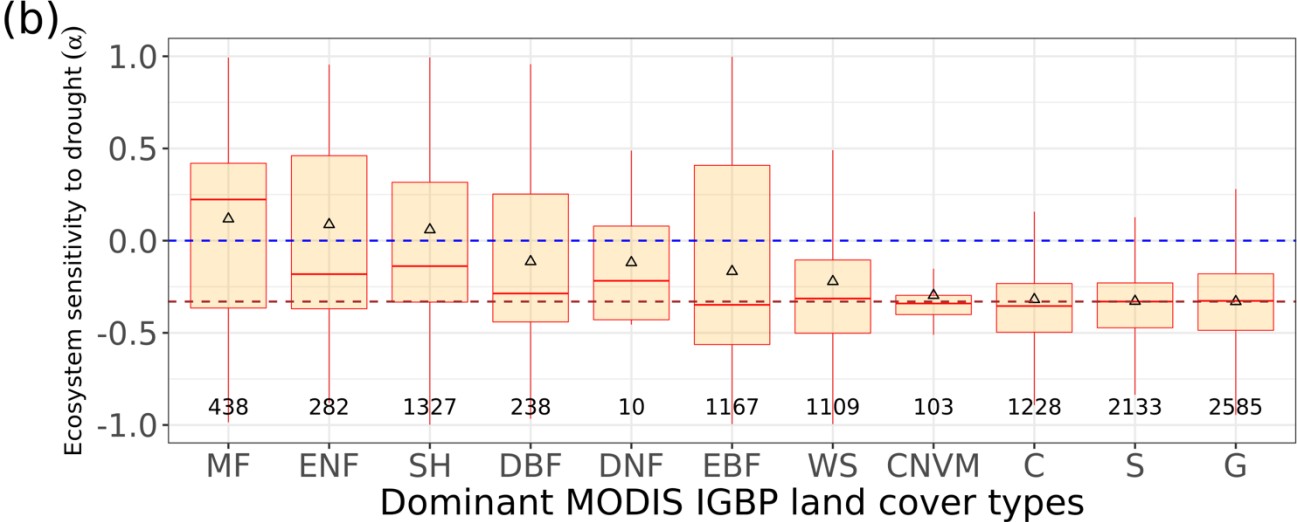

**Figure A3.** Distribution of ecosystem resistance $\alpha$ from (a) EVI and (b) kNDVI to drought for different dominant IGBP vegetation classes. Only significant $\alpha$ from the linear model is selected ($P$-value < 0.05) are shown.

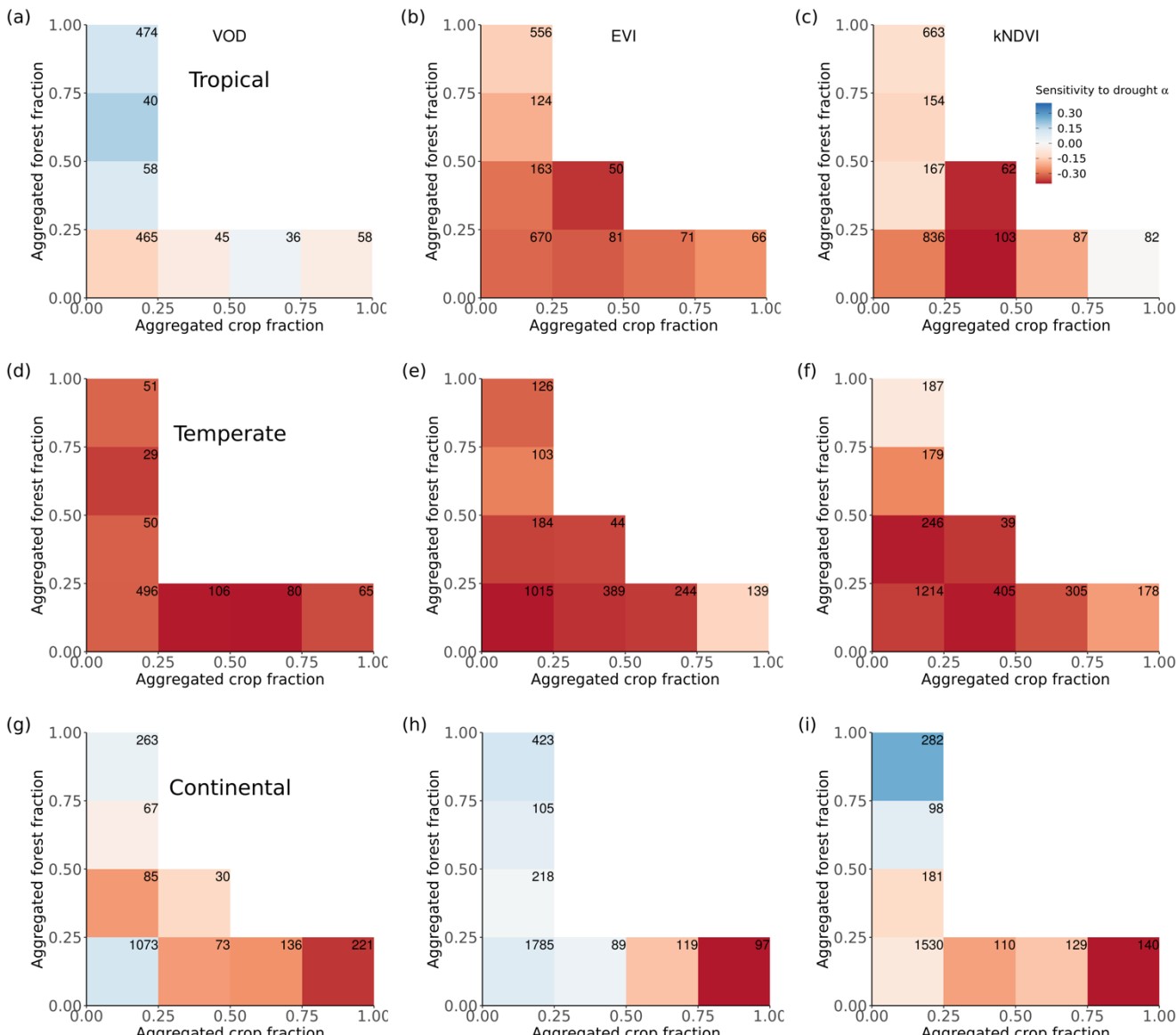

**Figure A4.** Ecosystem resistance to drought binned for different levels of the aggregated forest and cropland fraction classes from the three land cover products (a) L-VOD, (b) EVI and (c) kNDVI for Koeppen main climate class tropical climate and (d-f) for the temperate climate and (g-i) for the continental climate. Only significant coefficients $\alpha$ in the linear model ($P$-value < 0.05) are included and groups with less than 20 pixels are excluded.

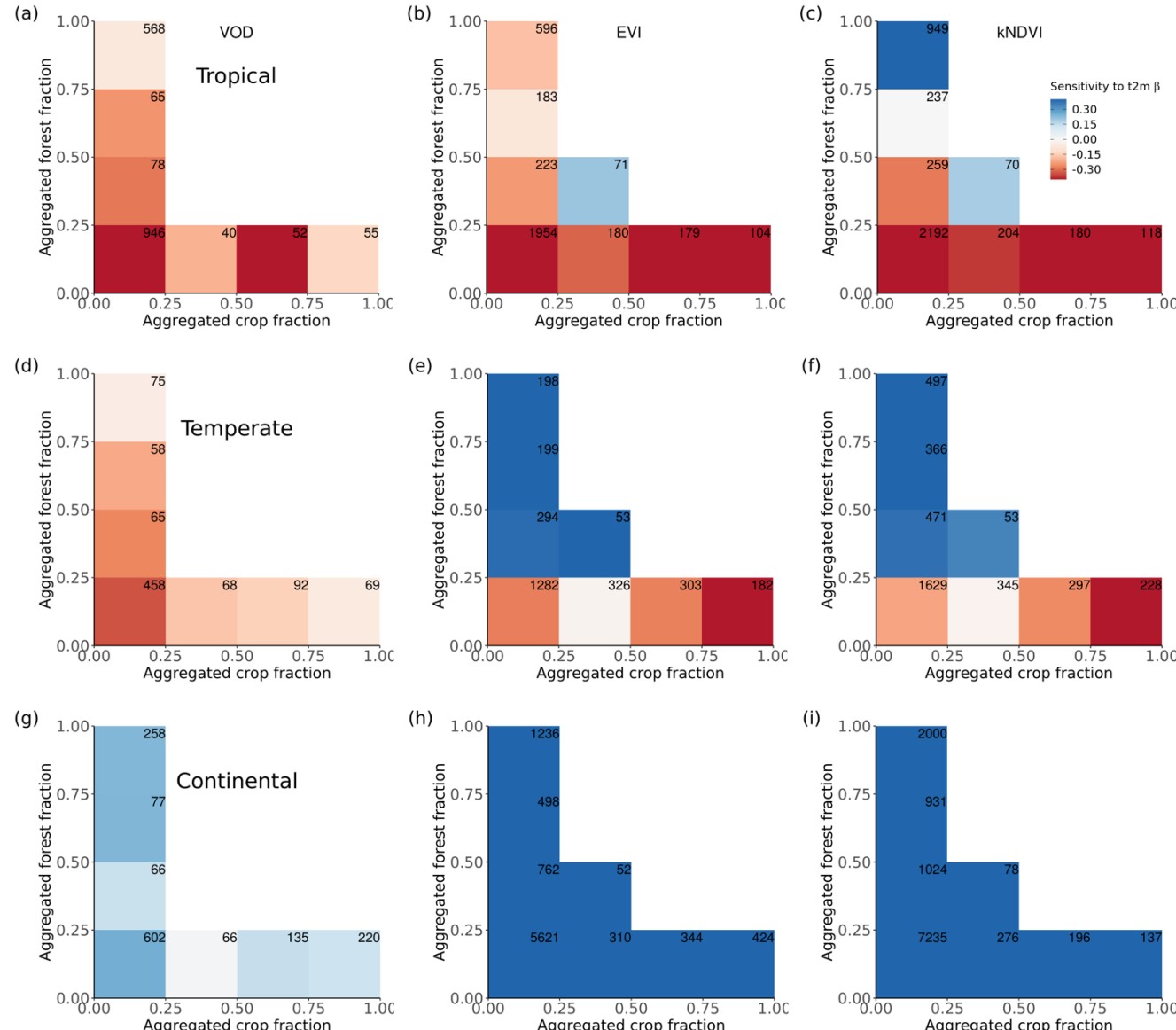

**Figure A5.** Ecosystem sensitivity to 2 m air temperature binned for different levels of the aggregated forest and cropland fraction classes from the three land cover products (a) L-VOD, (b) EVI and (c) kNDVI for Koeppen main climate class tropical climate and (d-f) for the temperate climate and (g-i) for the continental climate. Only significant coefficients $\beta$ in the linear model (*P*-value < 0.05) are included and groups with less than 20 pixels are excluded.

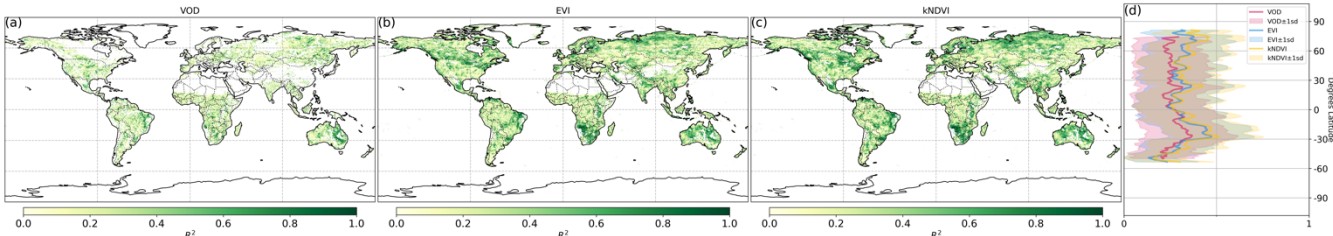

**Figure A6.** Ecosystem resistance to drought duration and its standard error. Spatial map of drought coefficients $\alpha$ for (a) L-VOD, (b) EVI, (c) kNDVI and their standard error (e, f, g). Same for temperature coefficients $\beta$ (i, j, k) and their standard error (m, n, o). The averages for different latitudes and their standard deviations are shown on the right (d, h, l, p).

**Figure A7.** Spatial map of $R^2$ for (a) L-VOD, (b) EVI, (c) kNDVI. The averages for different latitudes and their standard deviations are shown on the right (d).

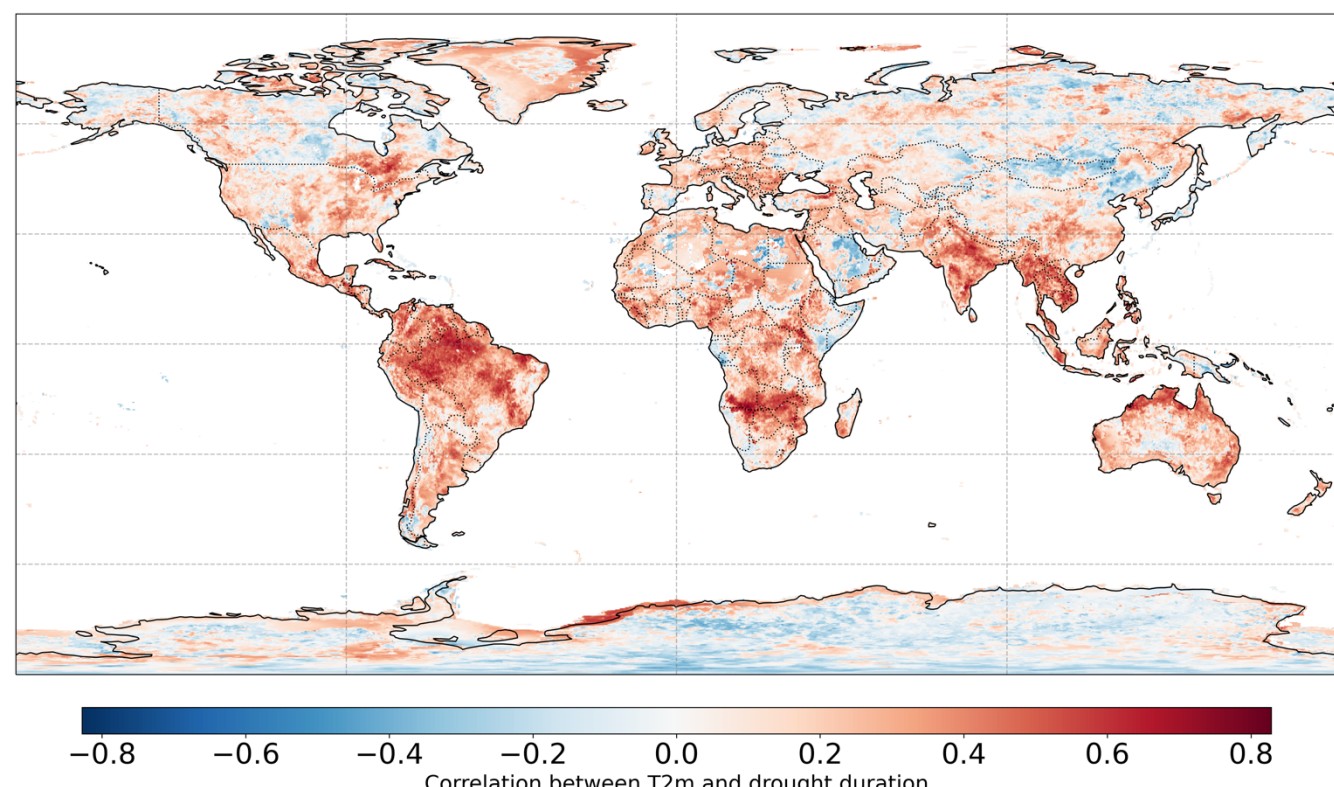

**Figure A8.** Temporal correlation between yearly mean temperature and drought duration (months per year) in 1979-2020.

**Table A1: IPCC AR6 reference sub-regions of land**

| Abbreviations | Full name | Abbreviations | Full name |
|---|---|---|---|
| GIC | Greenland/Iceland | SEAF | S.Eastern-Africa |
| NWN | N.W.North-America | WSAF | W.Southern-Africa |
| NEN | N.E.North-America | ESAF | E.Southern-Africa |
| WNA | W.North-America | MDG | Madagascar |
| CNA | C.North-America | RAR | Russian-Arctic |
| ENA | E.North-America | WSB | W.Siberia |
| NCA | N.Central-America | ESB | E.Siberia |
| SCA | S.Central-America | RFE | Russian-Far-East |
| NWS | N.W.South-America | WCA | W.C.Asia |
| NSA | N.South-America | ECA | E.C.Asia |
| NES | N.E.South-America | TIB | Tibetan-Plateau |
| SAM | South-American-Monsoon | EAS | E.Asia |
| SWS | S.W.South-America | ARP | Arabian-Peninsula |
| SES | S.E.South-America | SAS | S.Asia |
| SSA | S.South-America | NAU | N.Australia |
| NEU | N.Europe | CAU | C.Australia |

| | | | |
|---|---|---|---|
| WCE | West&Central-Europe | EAU | E.Australia |
| EEU | E.Europe | SAU | S.Australia |
| SAH | Sahara | NZ | New-Zealand |
| WAF | Western-Africa | EAN | E.Antarctica |
| CAF | Central-Africa | WAN | W.Antarctica |
| NEAF | N.Eastern-Africa | | |

**Appendix B: Comparison between linear models with and without the AR1 term as memory effects**

We tested the model form with a lag-1 autoregressive (AR1) term to consider the memory effects at an annual scale, which is the dependence of a vegetation state (i.e. L-VOD, EVI, kNDVI) anomaly on the previous year's anomaly. This term relates to the speed of the vegetation to return to its normal state and thus can be associated with the ecosystem resilience in an environment driven by climate factors dominantly (De Keersmaecker et al., 2016). This term is also interpreted as biological memory (Liu et al., 2019a), which may reflect intrinsic ecosystem feedbacks such as biomass accumulation or loss (Barron-Gafford et al. 2014). It also reflects the effects of other environmental and climate drivers or nonlinear responses not explicitly considered in our model (Liu et al., 2019a). Based on flux tower data, biological memory is found to contribute to increased $R^2$ on daily net ecosystem exchange (NEE) prediction (Cranko Page et al., 2022) and important to account for the above internal feedbacks, especially disturbances and human activities (Liu et al., 2019a). Therefore, we used the linear autoregressive model Eq. (2) shown in Line 226.

However, this memory effect might differ when we consider the different vegetation indices, and when we analyze at an annual scale. We have again tested whether adjusted $R^2$ generally increases when we include the lag-1 autoregressive term Y(t-1) compared to the model without this term. The spatial distribution of the differences in drought resistance $\alpha$, temperature sensitivity $\beta$ and adjusted $R^2$ is shown in Figure B1.

The overview of the differences is shown in Table B1. The averages of drought resistance differences $\Delta\alpha$ are close to 0. The spatial correlation between $\alpha$ calculated with/without memory term $\varphi$ is close to 1. The averages of temperature resistance $\Delta\beta$ differences are also close to 0. The spatial correlation between $\beta$ is also close to 1. The averages of the adjusted $R^2$ are all close to 0 and even negative, which means that introducing the memory term does not contribute to explain more variance of yearly vegetation state in 2011-2020. Because the drought resistance and temperature resistance are similar and we only have 10 year time series, to avoid overfitting, we have decided to use Eq. (2) without the AR1 term.

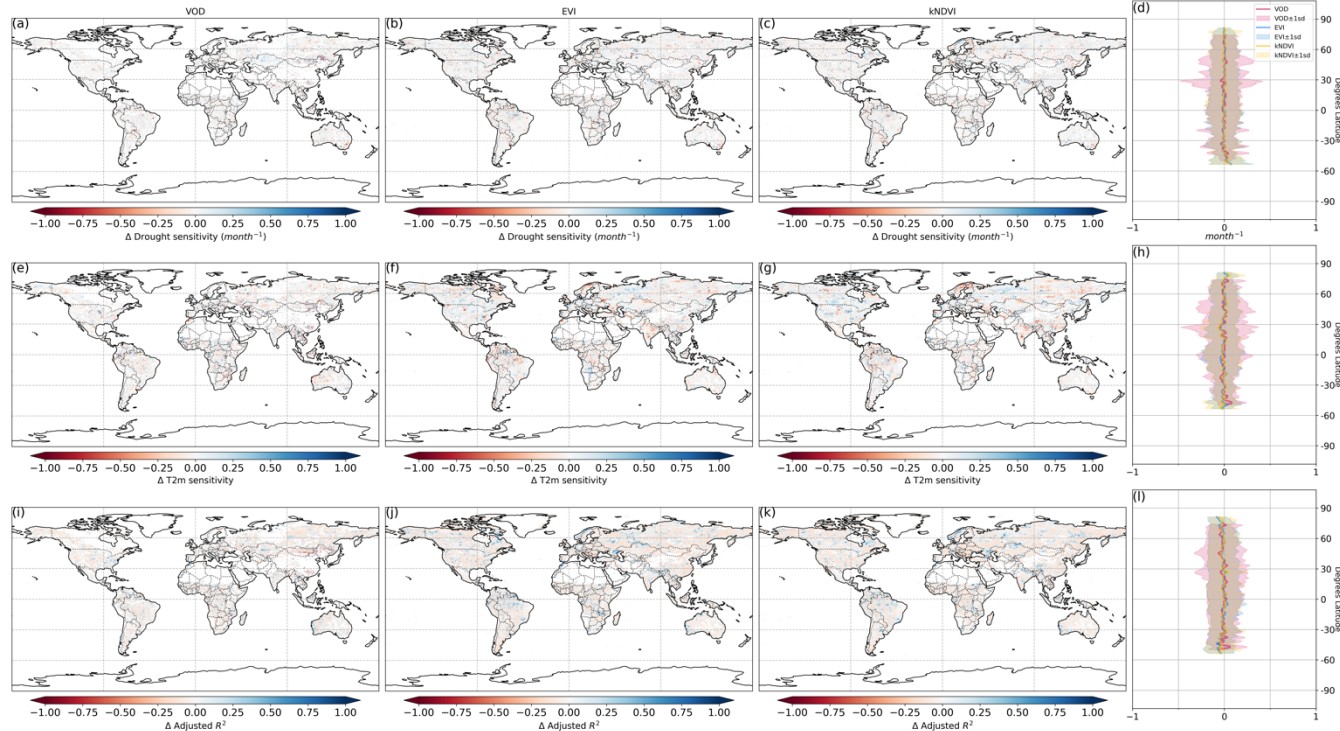

**Figure B1.** Difference between linear regression model with/without memory term (model with $\varphi$ minus model without $\varphi$) for ecosystem resistance to drought duration $\alpha$ from L-VOD, EVI and kNDVI (a, b, c). Similar for temperature sensitivity $\beta$ (e, f, g) and adjusted $R^2$ (i, j, k). The averages for different latitudes and their standard deviations are shown on the right (d, h, l).

**Table B1.** Overview of the differences in coefficients and adjusted $R^2$ between models with/without memory term $\varphi$

|  | L-VOD | EVI | kNDVI |
| --- | --- | --- | --- |
| Mean $\Delta\alpha$ (*month$^{-1}$*) | 0.0002 | -0.0013 | -0.0021 |
| Mean $\Delta\beta$ | -0.0064 | -0.0049 | -0.0023 |
| Mean $\Delta$adjusted $R^2$ | -0.017 | -0.012 | -0.014 |
| Spatial correlation ($\alpha$) | 0.88 | 0.93 | 0.93 |
| Spatial correlation ($\beta$) | 0.90 | 0.95 | 0.96 |

*Code and data availability*. The reconstructed SMOS L-VOD data are available upon request from Dr. Yang (huiyang@bgc-jena.mpg.de). MODIS EVI and NDVI data are freely available from https://modis.gsfc.nasa.gov/data/dataprod/mod13.php. ERA5 reanalysis data are freely available from https://www.ecmwf.int/en/forecasts/datasets/reanalysis-datasets/era5. MCD12Q1 land cover product is freely available from https://lpdaac.usgs.gov/products/mcd12q1v006/. ESA Land Cover CCI product is freely available from https://www.esa-landcover-cci.org/?q=node/164. LUH2 v2h land cover product is freely available from https://luh.umd.edu/data.shtml. The definitions of the AR6 reference sub-regions regions, the code and the spatially aggregated datasets are available at the GitHub ATLAS repository: https://github.com/SantanderMetGroup/ATLAS, https://doi.org/10.5281/zenodo.3998463 (Iturbide et al., 2020) under the Creative Commons Attribution (CC-BY) 4.0 licence. The global map of irrigation areas is freely available from https://www.fao.org/aquastat/ru/geospatial-information/global-maps-irrigated-areas/latest-version/. The global forest age map is freely available from https://www.bgc-jena.mpg.de/geodb/projects/Home.php. Select analysis codes used in this study are available at https://doi.org/10.5281/zenodo.8434669 (Xiao et al., 2023).

*Author contributions*. CX, AB, SZ designed the study. CX implemented the method and performed the data analyses. HY, JPW provided the L-VOD data. HY processed and reconstructed the L-VOD data. CX, AB, SZ prepared the first draft, and all authors commented on the paper and provided feedback throughout the data analysis.

*Competing interests*. The authors declare that they have no conflict of interest.

*Acknowledgements*. Chenwei Xiao acknowledges support from the International Max Planck Research School for Global Biogeochemical Cycles. We would like to thank Christiane Schmullius for helpful and stimulating discussions. We thank Nuno Carvalhais for initial discussions about the use of vegetation proxies. We thank Ulrich Weber for the preparation of ERA5 data, MODIS land cover, NDVI and EVI data.

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
