# Peer review of "Land cover and management effects of ecosystem resistance to drought stress"

_EGUsphere, 2023_

## Author Response (AR1)

Chenwei Xiao

Max Planck Institute for Biogeochemistry
Hans-Knöll-Str. 10
D-07745 Jena, Germany
cxiao@bgc-jena.mpg.de

Gabriele Messori
Editor-in-Chief
*Earth System Dynamics* journal

August 14, 2023

Dear Dr. Messori,

I am writing to submit the revision to our manuscript entitled "Land-cover and management modulation of ecosystem resistance to drought stress" for consideration for publication in Earth System Dynamics. We appreciate the constructive feedback from the reviewers, and we have taken their suggestions into consideration carefully during the revision.

We have responded to the reviewers in a separate point-by-point response document and included a revised version of the MS with tracked changes.

Please note that during the revision process, we included an additional co-author who had not been able to revise and approve the submitted manuscript (C. Schmullius). Therefore, we now include a revised list of co-authors, and we introduced changes in the text in light of the new co-author's comments. The core findings and conclusions have not been affected.

We have no conflicts of interest to disclose.

Thank you for your consideration of this manuscript. Please feel free to contact me if you have any questions or require further information.

Sincerely,

Chenwei Xiao, PhD
Max Planck Institute for Biogeochemistry
Hans-Knöll-Str. 10
D-07745 Jena, Germany
cxiao@bgc-jena.mpg.de

**Land-cover and management modulation of ecosystem resistance to drought stress**

Xiao, C., Zaehle, S., Yang, H., Wigneron, J.-P., Schmullius, C., and Bastos, A. *Earth Syst. Dynam. Discuss*

**Response to Reviewer #1**

**R1C1: This study examines vegetation sensitivity to droughts and heat on a global scale using several satellite-derived proxies of vegetation conditions, including L-VOD, EVI, and kNDVI. To estimate vegetation sensitivity, the authors propose an autoregressive model that incorporates annual drought frequency, annual thermal condition, and the previous year's vegetation states. By comparing the sensitivities to drought and heat across different land cover types, including primary and secondary forests, the authors identify distinct differences in vegetation sensitivities, which they term ecosystem resistance. The study highlights the role of forest cover and land management in shaping the ecosystem resistance to droughts and suggests the advantages of using L-VOD in monitoring vegetation dynamics for dense forests. The research addresses relevant scientific questions within the scope of ESD and provides a novel perspective that considers land cover differences, particularly with respect to the effects of land management practices such as forest management and irrigation. The scientific methods and assumptions are clearly outlined, and the results are well-presented.**

We thank the reviewer for the overall positive evaluation of our study and for the valuable insights. Below we offer a detailed point-by-point response to the comments.

**R1C2: One main concern about this study is the ambitious claim of "management modulation" of ecosystem resistance, which may not be straightforward to conclude. The results are based on linear autoregression models, and the management effects are derived from comparing primary and secondary forests that span multiple climate zones. It is possible that the differences observed between primary and secondary forests (e.g., Fig. 5a) are due to climate differences rather than forest management. For example, NEU and EEU, which are both dominated by secondary forests (Fig. 1; Fig. 3a), have contrasting alpha values (Fig. 3b) and different Köppen climate classifications. To better isolate the primary-secondary forest differences and see the effects of forest management, it would be helpful to exclude variations related to other drivers or to group Fig. 1 based on climate zones. Otherwise, the current conclusion that "primary forests, typically associated with higher biodiversity, tend to show stronger resistance to droughts than secondary forests" could be misleading, as the differences may simply be related to ecosystem types shaped by climate rather than forest management.**

We agree with the reviewer that a direct causal relationship cannot be fully evaluated by our analysis. We propose to change the word 'modulation' to 'effects' in our title and main text. See more details in our reply to R1C3.

**R1C3: The current title of the study seems ambitious when using the word "modulation." It may be more appropriate to use a different word, such as "differences," to accurately reflect the findings of the study. The word "modulation" suggests a strong causal relationship, but what we see here are simply differences between primary and secondary forests.**

We agree with the reviewer that the term 'modulation' may imply a strong direct causal relationship. Instead, we have changed the title to "Land cover and management effects on ecosystem resistance to drought stress". We used 'effects' to indicate that we have taken into account some effects from climate background, such as Koeppen climate classification, long-term temperature, and precipitation average, and we have provided possible explanations for the effects.

We would like to highlight that in the original manuscript, we have controlled for climate factors, namely climatological temperature and precipitation for comparison between primary and secondary forests. We only compared the pixels dominated by primary and secondary forests in a similar climate background defined by long-term mean temperature and precipitation. We had provided more details in the submitted manuscript Section 2.5 Lines 245-254. We acknowledge that we need to refer to the method again when analyzing our results. We would like to clarify this point by adding one sentence in our results section 3.3 now in Lines 358-360:

*It should be noted that to minimize the potential effects of the fact that primary and secondary forests are distributed differently across climate zones, we compare pairs of pixels with primary and secondary forests under similar temperature and precipitation climatological conditions bins as described in section 2.5.*

Following this approach, we detected a significant difference between primary and secondary forests as shown in Fig. 5a both in the submitted and updated versions without the AR1 term.

We have also accounted for the climate effects by comparing drought resistance for different forest and crop fractions in tropical, temperate, and boreal climate zones from Koeppen climate zones as shown in Fig. A4, Fig. A5, and discussed now in Lines 420-424.

In the submitted manuscript, we already tried to reduce the confounding effects due to climate by comparing only pixels with different forest ages in the tropics. We have limited the forest type effects by comparing only tropical evergreen broadleaf forests of different ages because only this forest type contains sufficient pixels spanning from 0 to 300 years old for a robust comparison. We realize that this is not clear in our methods, and now add the following sentences to Lines 259-260:

*To minimize the potential climate confounding effects on the dependence of α on forest age, we limited our comparison in the tropics due to limited pixels with ≥ 300 years old trees for other regions.*

Furthermore, to explain these effects, we have discussed some possible mechanisms to explain the stronger drought resistance in primary forests in Section 4.3, but we acknowledge that more explanation and supporting evidence are needed. We have added more discussion to Section 4.3, now in Lines 447-458:

*Primary forests also show higher hydraulic diversity than secondary forests, which buffers impacts in ecosystem flux during dry periods across temperate and boreal forests (Anderegg et al., 2018). Secondary forests in the Brazilian Amazon are vulnerable to drought stress with a lower carbon balance and growth rates, and they only reached 56% of the tree diversity in the nearest primary forests (Elias et al., 2020). In Amazon forests, forest greening in degraded forests disturbed by fire has been found to be more dependent on water resources than in mature forests (Roux et al., 2022). The new forest edges in much more fragmented degraded forest landscapes increase canopy desiccation, tree mortality, and fire frequency (Briant et al., 2010; Broadbent et al., 2008), especially during drought events (Roux et al., 2022).  In boreal forest ecosystems in Sweden, primary forests have been found to be less affected by drought compared to secondary forests (Wolf et al., 2023). Primary forests also likely harbor older trees, which also show higher resistance to drought (Fig. 5b). Besides, primary forests might have a more extensive rooting system with higher availability of soil water. However, it remains difficult to disentangle the above factors for the complex ecosystem due to limited data.*

We also changed 'modulation' in Lines 23-24 to:

*We analyze how ecosystem resistance varies with land cover across the globe and investigate the potential effects of forest management and crop irrigation.*

Lines 441:

***4.3 Effects of land management and forest age on ecosystem resistance***

Lines 460:

*Apart from the effect of land cover due to different sensitivity to drought stress for different vegetation types, ...*

Line 510:

*The effects of specific land cover and land management on drought resistance are thus important for these regions in the future.*

**Comments on the analytical methods**

**R1C4: I have several questions regarding the linear autoregressive model used in this study. First, how is it ensured that the droughts used in the first term occur within or before the growing season and affect vegetation growth? Second, is there a justification for using yearly mean temperature instead of yearly maximum temperature in the second term? It would be helpful to either provide relevant references or explain the advantages of using annual mean temperature (e.g., smaller prediction errors compared to using yearly maximum). Third, the third term in the model is incorporated to consider vegetation memory effects, but the study does not present any results related to this coefficient. It would be helpful to briefly mention any relevant findings even if they are not significant. Additionally, it's worth questioning the inclusion of this term in the model if it does not contribute to reducing the overall prediction error. Fourth, for better readability, it would be helpful to mention the "c" term in section 2.4 of the study.**

Thanks for the constructive comments on our analysis methods. We address the four points below, (i) drought in the growing season; (ii) temperature predictor; (iii) AR1 term; (iv) description for the intercept term c.

(i) First, we acknowledge that drought months might not exactly overlap with the growing season. We analyzed yearly maximum L-VOD, which normally happens in the growing season and yearly mean EVI, kNDVI which reflect the vegetation conditions during the whole year but are mostly affected by growing season values.

Drought occurring before the growing season will be reflected in the yearly mean EVI and kNDVI. Intense drought happening prior to the growing season is also likely to have legacy effects on vegetation growth during the growing season (e.g. Buermann et al., 2018; Bastos et al., 2020). Therefore, the sub-annual effects of drought prior to or during the growing season are expected to be reflected in the yearly maximum L-VOD. Disturbance prior to the growing season also directly leads to decreased yearly maximum values during the growing season. Droughts occurring after the growing season, however, would not have any effect on the yearly maximum L-VOD as estimated here. They would affect mean annual EVI and kNDVI but should have a small contribution since the impact would be expressed during the senescent period. We do not consider explicitly interannual legacy effects from droughts, but we have performed a sensitivity analysis on the effects of interannual long-term memory effects, which did not affect the results (now moved to Appendix B).

To evaluate whether the non-growing season drought might influence the results, we selected only drought months occurring during the growing season (April to September) in the Northern Hemisphere extratropics, where vegetation shows clear seasonality. The results are

shown in Fig. R1. The spatial pattern of negative and positive values of drought resistance is not significantly changed, with $\alpha$ calculated from growing season drought ($\alpha_{gs}$) and $\alpha$ calculated from the whole year drought ($\alpha_{year}$) showing a spatial correlation of 0.80 (Fig. R1a, b, c). The averaged difference in the multiple linear regression model $R^2$ is 0.003 and the median is 0. The absolute values of $\alpha_{gs}$ become higher in some areas because of the shorter drought duration when considering only the growing season drought (Fig. R3). In 75 pixels, the absolute values of $\alpha_{year}$ decrease from values above 1 to values below 0.1 of $\alpha_{gs}$ as shown by the values close to the zero line in Fig. R3. The values of $\alpha_{gs}$ in those 75 pixels are all insignificant ($P$-value $\geq 0.05$). The reason for such differences could be partly due to a much smaller annual variance of drought duration in the growing season in these few pixels (N = 75) for 2010-2020 (0.96 in growing season drought duration, 2.89 in the whole year drought duration), which leads to higher standard errors and unreliable estimates of $\alpha_{gs}$. When the drought duration within the growing season exactly matches the whole year drought duration, the calculated drought resistance is the same (close to the 1:1 line in Fig. R3). Overall, our main results for the differences between forest and crops, and effects of land management practices still hold. The spatial patterns of temperature sensitivity for L-VOD are also similar, with a spatial correlation of 0.96 (Fig. R2). Similar results are found in EVI and kNDVI (not shown).

[Figure]

**Figure R1.** Comparisons between the linear autoregressive model with the whole year drought or growing season drought duration for ecosystem resistance to drought duration $\alpha_{year}$ (a) and $\alpha_{gs}$ (b) from L-VOD. The differences in $\alpha$ and $R^2$ (model with whole year drought minus growing season are shown in (c, d).

[Figure]

**Figure R2.** Comparisons between the linear autoregressive model with the whole year drought or growing season drought duration for temperature sensitivity $\beta$ (a, b) from L-VOD. The differences in $\beta$ and $R^2$ (model with whole year drought minus growing season) are shown in (c, d).

[Figure]

**Figure R3.** The comparison between ecosystem resistance to the whole year drought duration and growing season drought duration.

(ii) Second, we agree that there is an inconsistency between the yearly mean temperature and the discussion about heat stress. We include yearly mean temperature in our linear regression model due to the following two arguments:

First, yearly mean temperature strongly correlates with drought duration (months per year) in 1979-2020 (Fig. R4). This is especially marked for tropical and semi-arid regions, with correlations between the two variables over 0.6. Therefore, if a simple linear regression model with drought duration as the only predictor would be used, the coefficients would implicitly include the effects of mean annual temperature variations. To estimate drought resistance more reliably, one needs to control for temperature, justifying the use of a more complex model.

[Figure]

**Figure R4.** Temporal correlation between yearly mean temperature and drought duration (months per year) in 1979-2020.

Second, after controlling for temperature, the adjusted $R^2$ of the model increases by more than 0.1 in 26% and 40%, and 44% of the pixels for L-VOD, EVI, and kNDVI. This justifies fitting a model with two degrees of freedom, versus a simple linear regression.

Based on the above arguments, we have now changed it to "temperature sensitivity".

We changed the sentence in Line 229 now to:
*Similarly, we used β as a metric for the ecosystem sensitivity to temperature anomalies.*

Lines 231-232 to:
*α and β were evaluated for each pixel.*

(iii) Third, we added the AR1 term to consider the memory effects at an annual scale, which is the dependence of a vegetation state (i.e. L-VOD, EVI, kNDVI) anomaly on the previous year's anomaly. The AR1 term expresses the speed of the vegetation to return to its normal state following any perturbation (i.e. not necessarily drought-related). This term is often associated with ecosystem resilience in an environment driven by climate factors dominantly (De Keersmaecker et al., 2016). This term can also be interpreted as biological memory (Liu et al., 2019), which may reflect intrinsic ecosystem feedback such as biomass accumulation or loss (Barron-Gafford et al. 2014) as well as effects of other environmental and climate

drivers or nonlinear responses not explicitly considered in linear models such as the one used here (Liu et al., 2019). Based on flux tower data, biological memory has been found to contribute to increased $R^2$ on daily net ecosystem exchange (NEE) prediction (Cranko Page et al., 2022) and to vary with the above internal feedbacks, especially disturbances and human activities (Liu et al., 2019). Therefore, we initially applied the linear autoregressive model Eq. (2) shown in Line 184 in the initial submitted manuscript.

However, such memory effects might differ when we consider the different vegetation indices at annual scales. To evaluate whether the AR1 term carries additional information to the variability in our target variables, we compared the adjusted $R^2$ of the model with and without the AR1 term. Generally, the adjusted R2 increases when we include the AR1 term. The spatial distribution of the differences in drought resistance $\alpha$, temperature sensitivity $\beta$, and adjusted $R^2$ is shown in Fig. R5.

[Figure]

**Figure R5.** Difference between linear regression model with/without memory term (model with $\varphi$ minus model without $\varphi$) for ecosystem resistance to drought duration $\alpha$ from L-VOD, EVI and kNDVI (a, b, c). Similar for temperature sensitivity $\beta$ (e, f, g) and adjusted $R^2$ (i, j, k). The averages for different latitudes and their standard deviations are shown on the right (d, h, l).

The overview of the differences is shown in Table R1. The average values of the difference in $\alpha$ ($\Delta\alpha$) and $\beta$ ($\Delta\beta$) are close to 0. The spatial correlation between $\alpha$ and $\beta$ calculated with/without memory term $\varphi$ is 0.88 or higher. The averaged $\Delta$adjusted $R^2$ are all close to 0 and sometimes even negative, which means that introducing the memory term does not contribute to explaining more variance of yearly vegetation state in 2011-2020.

Table R1. Overview of the differences in coefficients and adjusted $R^2$, and spatial correlation of $\alpha$ and $\beta$ between models with/without memory term $\varphi$

|  | L-VOD | EVI | kNDVI |
|---|---|---|---|
| Mean $\Delta\alpha$ (month$^{-1}$) | 0.0002 | -0.0013 | -0.0021 |
| Mean $\Delta\beta$ | -0.0064 | -0.0049 | -0.0023 |
| Mean $\Delta$adjusted $R^2$ | -0.017 | -0.012 | -0.014 |
| Spatial correlation ($\alpha$) | 0.88 | 0.93 | 0.93 |
| Spatial correlation ($\beta$) | 0.90 | 0.95 | 0.96 |

Because the drought resistance and temperature sensitivity are similar and we only have a 10-year long time series, to avoid overfitting, we have decided to remove the memory term in new Eq (2), now in Lines 205-206:

*To calculate the ecosystem resistance to drought, we applied a linear model for each pixel following Eq. (2):*
$$Y_{anom}(t) = \alpha N(t) + \beta T_{anom}(t) + c + \epsilon(t) \tag{2}$$

and removed the description for $\varphi$ in Lines 230 now from:

"*The third term of the equation corresponding to the previous year's vegetation anomalies could be associated with the recovery rate to the average state or persistent impact during the next year. Thus, $\varphi$ can represent vegetation memory, which has previously been proposed to be associated with long-term resilience to any type of perturbation. $\alpha$ is in the unit of month$^{-1}$ but $\beta$ and $\varphi$ have no unit due to standardization.*"

to:

*$\alpha$ is in the unit of month$^{-1}$ but $\beta$ has no unit due to standardization.*

and we discussed the reason to choose to use a more simple model in Lines 215-222 and Appendix B:

*We also analyzed the linear autoregressive model as Eq. (4):*
$$Y_{anom}(t) = \alpha N(t) + \beta T_{anom}(t) + \varphi Y_{anom}(t-1) + c + \epsilon(t) \tag{4}$$

*in which we also considered the memory effect with a lag-1 autoregressive (AR1) term, which has been demonstrated to be fundamental to understanding the daily carbon metabolism of terrestrial ecosystems (Liu et al., 2019; Cranko Page et al., 2022) and*

*ecosystem resilience of monthly NDVI (De Keersmaecker et al., 2016). We evaluated if interannual memory effects influence drought sensitivity, temperature sensitivity, and adjusted $R^2$, but the additional predictor does not carry new information for annual L-VOD, EVI, and kNDVI anomalies. The drought sensitivity and temperature sensitivity are similar and adjusted $R^2$ does not increase in general (Appendix B), so we used the most parsimonious model Eq. (2).*

and we have changed the sentence describing the method in our abstract in Lines 21-23 to:

*We apply a linear model accounting for drought and temperature effects to characterize ecosystem resistance by their sensitivity to drought duration and temperature anomalies.*

We also changed the caption for Fig. 3 in Line 312-315 now to:

*(b) Distribution of ecosystem resistance α to drought for different reference sub-regions, the number in each box is the number of pixels in this category. Only regions with averages that are statistically different from zero are shown (two-sided Student t-test; P-value < 0.05); (c) Distribution of ecosystem resistance to drought α for different dominant IGBP vegetation classes. Only significant α from the linear model is selected (P-value < 0.05) are selected.*

Similarly, we substitute the "linear AR1 model" with the "linear model" in our main text and figure captions.

We have moved the discussion about the AR1 term, Fig. R5 and Table R1 in Appendix B.

We have also updated all results and figures based on the new Eq. (2). The main results and conclusions of the original study do not change.

There are some small changes in our results:

Lines 268-269:
*At the global scale, α based on Eq. (2) are negative in 55% of pixels with valid values from L-VOD, 56% from EVI, and 59% from kNDVI (Figure A6a-A6c, Figure 3a).*

Lines 271-274:
*but in Southeast Asia, EVI shows the lowest drought resistance with a median of -0.12 month$^{-1}$ in disagreement with L-VOD which has a median of 0.03 month$^{-1}$. kNDVI shows the same sign as EVI, with a median of -0.05 month$^{-1}$. In the Amazon, there are around 15% and 18% more pixels showing negative resistance from EVI and kNDVI than L-VOD.*

Lines 295-298:

32 regions coloured red or blue in Fig. 3a have mean values of $\alpha$ significantly different from zero, as determined by the two-tailed Student t-test (P-value < 0.05). The median values of $\alpha$ range from -0.20 to 0.07 month$^{-1}$ across regions, and the average values from -0.25 to 0.08 month$^{-1}$. Among these regions, 25 regions show significant negative mean $\alpha$.

Lines 301:

In the tropics, SAS, CAF, WAF, and NAU, which have lower forest cover show negative mean $\alpha$, ...

Lines 304-307:

$\alpha$ based on L-VOD are higher in evergreen needleleaf forests (ENF), shrublands (SH) and evergreen broadleaf forests (EBF), but lower in cropland (C), deciduous needleleaf forests (DNF), and crop/natural vegetation mosaic (CNVM) (Figure 3c). kNDVI and EVI agree on the lower resistance in C and CNVM and high resistance in SH, ENF, and mixed forests (MF) (Figure A3).

Lines 320-324:

Pixels dominated by forests are significantly (P-value < 0.05; indicated by stars) more resistant to droughts than those where cropland predominates (Figure 4a-c). $\alpha$ increases with increased forest fraction, from a mean value of -0.07 month$^{-1}$ (0-25% forest cover) to 0.07 month$^{-1}$ (75-100% forest cover), and decreases with increased cropland fraction, from -0.07 month$^{-1}$ for 0-25% crop cover to -0.30 month$^{-1}$ for 75-100% crop cover.

Lines 362-366:

The median of the difference between primary and secondary forests ($\Delta\alpha$) is 0.382 month$^{-1}$ and is significantly greater than 0 based on the one-sample Wilcoxon test (P-value < 0.05). However, we did not detect such a large difference between EVI and kNDVI, whose medians of $\Delta\alpha$ between forest types are -0.040 month$^{-1}$ and 0.052 month$^{-1}$, but given the large spread of the distribution, their medians are not significantly greater than 0 based on the one-sample Wilcoxon test (P-value > 0.05).

Lines 370-373:

The median of $\alpha$ for forests younger than 100 years is -0.458 month$^{-1}$ but the medians of $\alpha$ of forests from 100 to 300 years and forests older than 300 years are 0.429 month$^{-1}$ and 0.361 month$^{-1}$ respectively. We also find a significant (P-value < 0.05) increase of $\alpha$ in kNDVI between forests younger than 100 years and older than 300 years, but the effect is not as large as in L-VOD, which confirms that VOD is more sensitive to water volume and biomass than kNDVI in general.

Lines 377-381:

We finally investigated the ecosystem resistance $\alpha$ for different irrigation levels (Figure 5c). The result shows also an increasing resistance for L-VOD with the irrigation levels, with the median of $\alpha$ for L-VOD increasing from -0.342 month$^{-1}$ to 0.023 month$^{-1}$

*between less than 10% actually irrigated cropland and more than 50% actually irrigated cropland (P-value < 0.05). For kNDVI and EVI, the change in the median of α is negligible but we still found a higher percentage of pixels with close to zero or positive resistance for higher irrigation fractions.*

(iv) Fourth, we have added the description describing the term "c" in Line 210 now, thank you for the suggestion.

> *... c is the intercept term and ε is the residual term…*

**R1C5: One concern I have is about the explanatory power of each regression over each grid point, given that only 10 values are available for the regression. This could touch a pragmatic lower bound for sample size, and it is therefore important to ensure that the derived coefficients are significant and that their spatial patterns are indicated. For example, in Fig. 2, it would be helpful to show the significance of the derived coefficients along with their spatial patterns. This would allow readers to better assess the reliability of the results and understand the degree of confidence that can be placed in the findings.**

Thanks for suggesting showing the significant coefficients explicitly. To address this concern, we propose a new version of Fig. 2 to overlap the original maps with stippling and grey shade where results are NOT significant at a 10% significance level (reproduced below as Fig. R6). We also moved the original Fig. 2 to Appendix A as Fig. A6 to show the overall spatial distribution of drought resistance and temperature sensitivity and the standard error, which we described in our main text to indicate uncertainty now in Lines 275-277 and 286-287.

[Figure]

**Figure R6.** Ecosystem resistance to drought duration and temperature sensitivity. Spatial map of drought resistance α for (a) L-VOD, (b) EVI, (c) kNDVI. Same for temperature sensitivity β (e, f, g). The averages for different latitudes and their standard deviations are shown on the

right (d, h). The pixels with NON-significant $\alpha$ and $\beta$ at a 10% significance level are marked with stippling and plotted with a light grey shade to better show the significant results.

[Figure]

**Figure A6.** Ecosystem resistance to drought duration and its standard error. Spatial map of drought resistance $\alpha$ for (a) L-VOD, (b) EVI, (c) kNDVI and their standard error (e, f, g). Same for temperature sensitivity $\beta$ (i, j, k) and their standard error (m, n, o). The averages for different latitudes and their standard deviations are shown on the right (d, h, l, p).

**Ln 150: Are the land cover classification considering temporal changes? For example, land cover A for year 1 but becomes land cover B for year 2, what would be the eventual land cover for analysis?**

Thanks for pointing this out. Indeed, we have considered the temporal changes of land cover over 2010-2020 by only analyzing those pixels without dominant land cover classification changes for Fig. 3a, c. We realize that we failed to explain this, and now add the following sentences to Section 2.5 Line 237-238:

*We only analyzed those pixels without changes in the dominant IGBP vegetation cover class during the period 2010-2020.*

For Fig. 4, we selected only those pixels without forest, crop, and grass changes at a 25% interval level over 2010-2020. This was described in Section 2.3 Lines 182-183. This information is now added to the figure caption as well in Line 345-346:

*Only pixels with no change in the 25% bins of the four dominant vegetation categories (forests, shrublands, grasslands, and croplands) are analyzed.*

For Fig. 5, we realized that we failed to explain this, and now add the following sentences to Section 2.5 Line 238-241:

*We only analyzed those pixels with less than 10% changes in primary and secondary forest, tropical evergreen broadleaf forests, and crop cover fraction in 2010-2020. We used 10% to avoid abrupt and substantial changes in vegetation cover that might directly modulate the above-ground vegetation biomass variations and therefore affect our regression.*

We change the following sentence in Line 247 to better define the dominant primary forests and secondary forests:

'*First, the grid cells were filtered by > 50% forest fraction and those containing LUH2 v2h forest management information were categorized according to ERA5 temperature and total precipitation long-term averages.*' to

*First, we define primary forests as those pixels with > 50% forest fraction and primary forests dominate > 50% of the forest fraction, similar for secondary forests where secondary forests dominate > 50% of the forest fraction in 2010-2020. These pixels were then categorized according to ERA5 temperature and total precipitation long-term averages.*

For forest ages and crop irrigation, we add the following sentence in Line 258:
*We only selected pixels with over 50% forest fraction in 2011-2020 and no variation in the dominant vegetation type (tropical evergreen broadleaf forests).*

and in Lines 260-261:

*We only compare pixels where crop fraction is > 50% in 2011-2020 for irrigation effects.*

and figure caption as well in Line 389-390:

*Only pixels with unchanged dominant primary and secondary forest, tropical EBF, and crop cover in 2011-2020 were selected, as defined in Section 2.5.*

We add the description about how we determine the significance of the difference between two bins in our method Section 2.5 in Lines 263-264 now:

> *For the comparison between two groups, we applied the unpaired two-sample Wilcoxon test to test whether there is a significant difference between their medians (P-value < 0.05).*

We then make our results more concise without explicitly showing the test method we use: In Lines 371-373:

> *We also find a significant (P-value < 0.05) increase of α in kNDVI between forests younger than 100 years and older than 300 years, but the effect is not as large as in L-VOD, which confirms that VOD is more sensitive to water volume and biomass than kNDVI in general.*

In Lines 380:

> *...significantly (P-value < 0.05)...*

**Ln 188: It is not clear how the anomalies were standardized.**

We used the standard score of the anomalies and have now added a mathematical equation to better clarify our standardization methods (now Eq. 3) in Line 212:

$$\frac{X - \mu}{\sigma} \tag{3}$$

**Other comments**

**Ln 26: Up on the improved analysis of primary-secondary forest differences in alpha, this sentence "L-VOD indicates that primary forests tend to be more resistant to drought events than secondary forests" may need to be rephrased.**

We have rephrased the sentence in Lines 28-29 to:

> *L-VOD indicates that primary forests tend to be more resistant to drought events than secondary forests when controlling for the differences in background climate, but ...*

**Ln 27: "EVI and kNDVI saturation in dense forests.", do you mean for the biomass estimates? Note that EVI is designed to be less susceptible to saturation over dense forest areas (Huete et al., 2002: 10.1016/S0034-4257(02)00096-2).**

We agree with the reviewer that EVI is designed to be less susceptible to saturation than NDVI over forest areas for green vegetation and forest cover. However, reflectance-based indices still reflect mostly the top of canopy density/cover. For aboveground biomass

estimates, optical indices (i.e. NDVI or EVI) have been found to saturate in dense vegetation at moderate L-VOD values (Li et al., 2021).

**Ln 39: any reference for the concept "ecosystem resistance"?**

Yes, Gessler et al. (2020) extensively reviewed the concept of resistance in their study. We have restructured the sentence in Lines 42-46 to be:

> *..., i.e., the ecosystem resistance (Gessler et al., 2020; Ingrisch and Bahn, 2018) and recovery trajectory following the disturbance (Schwalm et al., 2017; Wigneron et al., 2020; Wu et al., 2022; Yao et al., 2023). Ecosystem resistance is defined as the concurrent impact of a disturbance on response parameters. However, event-specific resistance may differ under different drought conditions. We applied a long-term resistance definition as the vegetation sensitivity to drought duration over multiple years, making it consistent for spatial comparison.*

**Ln 40: Studies related to vegetation recovery and legacy effects have been increasing recently, more latest references are needed for supporting the sentence "recovery trajectory following the disturbance".**

We have added more recent references in Line 43:
> *... the recovery trajectory following the disturbance (Schwalm et al., 2017; Wigneron et al., 2020; Wu et al., 2022; Yao et al., 2023)*

**Ln 41: "The mitigation of climate extreme events and maintenance of land carbon sink are highly dependent on the resistance of ecosystems and their changes under land use and land cover change.": Looks a bit abrupt to come to this sentence, some transition may be needed. Also, please provide references for this sentence.**

Thanks for pointing out the abrupt transition. We have restructured this part and added the following sentences to Lines 46-51:
> *... Therefore, gaining a thorough understanding of the global patterns of ecosystem resistance and identifying the potential drivers behind their spatial variations is crucial for comprehending the impact of drought events on the land carbon sink. Potential drivers include climate background, plant species, biodiversity, tree height and ages and land use and land management. Land use and land management can potentially affect ecosystem resistance through changes in species composition, biodiversity and stand structure.*

**Ln 55-56: "Taller tropical forests … because …. ": note that the influence of tree height on the response of tropical forests to drought and subsequent non-drought growth remains controversial. The deep roots of the tropical forest may also play a critical role, check the studies Brando, 2018: https://doi.org/10.1038/s41561-018-0147-z and Giardina et al., 2018: https://doi.org/10.1038/s41561-018-0133-5.**

Thanks for pointing out and suggesting additional literature. We have rephrased this sentence in Lines 65-70 for clarity:

*Taller tropical forests have been shown to be more vulnerable to vapor pressure deficit (VPD) fluctuations during drought periods because of smaller xylem-transport safety margins, in studies based on Ku-band VOD (Liu et al., 2021) and SIF (Giardina et al., 2018). However, the opposite has been found when analyzing resistance to precipitation variability (Giardina et al., 2018) and terrestrial water storage anomalies (Liu et al., 2021), with taller trees being associated with higher resistance to soil-moisture deficit, based on the same vegetation proxies. However, some satellite products are less capable of detecting vegetation dynamics in dense forests.*

**Ln 58: "kNDVI is better correlated …", compared to which indices?**

We have rephrased this sentence in Lines 71-72 for clarity.

*kNDVI is better correlated with key vegetation parameters, such as leaf area index (LAI), gross primary productivity (GPP), and sun-induced chlorophyll fluorescence (SIF) compared to NDVI.*

**Ln 63: DVGMs or DGVMs?**

Corrected to DGVMs.

**Ln: 64-66: "DVGMs and upscaled FLUXCOM GPP have suggested that GPP anomalies are less negative or even positive for pixels including …": less negative than what? The entire sentence is a bit difficult to understand, good to rephrase it or divide it into several short sentences.**

Thanks for pointing this out. We have rephrased this sentence in Lines 77-83:

*For example in Europe, both DGVMs and upscaled FLUXCOM GPP have suggested that GPP anomalies under low soil moisture conditions are less negative, or even positive, in pixels with more than 80% forest cover than in those pixels with lower forest cover. On the contrary, pixels with higher crop cover show stronger negative impacts on GPP anomalies compared to the pixels with low crop coverage in drought and heat events that occurred in 2003, 2010 and 2018 in Europe (Bastos et al., 2020). Over the globe, FLUXCOM GPP shows a reduced sensitivity to depleted soil moisture with increased tree cover (Walther et al., 2019).*

**Ln 66: "However, ecosystem fluxes are not directly observable at the ecosystem scale." the definition of an ecosystem is quite broad and may be good to indicate what the ecosystem scale is referring to here.**

Thanks for pointing this out. We have already changed the sentence in Lines 83-84 to:

*However, ecosystem fluxes are not directly observable at a regional scale beyond the footprint of flux towers (a few hundred meters).*

**Ln 72-74: "For example, modifying forest density and structure by high-intensity overstory removal was tested in conifer-broadleaf mixed forests in Central Europe and considerably increased their growth resilience to droughts and decreased drought-induced mortality by two-thirds (Zamora-Pereira et al., 2021).": It could be helpful to indicate the findings mentioned here are based on a stand-alone forest gap model.**

Thanks, we agree. We have rephrased this sentence in Lines 90-92 to:

*For example, based on a stand-alone forest gap model, modifying forest density and structure by high-intensity overstory removal in conifer-broadleaf mixed forests in Central Europe considerably increased their growth resilience to droughts and decreased drought-induced mortality by two-thirds (Zamora-Pereira et al., 2021).*

**Ln 101: "VOD has been used 100 as a proxy for biomass", I guess you meant "aboveground" biomass.**

We have rephrased it to aboveground biomass.

**Ln 162: Table 3 can not be found.**

We apologize for the missing table. We have now included it in the main text after Line 185.

**Table 3. Overview of the merged input classes for the four vegetation categories used in this study.**

| Vegetation categories | MCD12Q1 | Land Cover CCI | LUH2 v2h |
|---|---|---|---|

| Forests | Evergreen Needleleaf Forests, Evergreen Broadleaf Forests, Deciduous Needleleaf Forests, Deciduous Broadleaf Forests, Mixed Forests | Tree Broadleaf Evergreen, Tree Broadleaf Deciduous, Tree Needleleaf Evergreen, Tree Needleleaf Deciduous | Forested primary land, Potentially forested secondary land |
|---|---|---|---|
| Shrublands | Closed Shrublands, Open Shrublands, Woody Savannas, Savannas | Shrub Broadleaf Evergreen, Shrub Broadleaf Deciduous, Shrub Needleleaf Evergreen, Shrub Needleleaf Deciduous | |
| Grasslands | Grasslands | Natural Grass | Managed pasture, Rangeland |
| Croplands | Croplands, Cropland/Natural Vegetation Mosaics | Crops | C3 annual crops, C3 perennial crops, C4 annual crops, C4 perennial crops, C3 nitrogen-fixing crops |

**Ln 180: Figure 1, what does the white area represent?**

The white area represents either (1) areas that are not dominated by primary or secondary forests, which are defined by pixels with > 50% forest fraction and > 50% of the forest fraction dominated by primary or secondary forests in 2011-2020, (2) where the irrigation ratio is zero or (3) where the minimum cropland fraction in MODIS is less than 50% in 2011-2020.

To make it consistent with our analysis of the effect of irrigation on drought resistance, we have modified Fig. 1 and changed the irrigation area ratio of cropland for minimum MODIS cropland > 25% to > 50% in 2011-2020 (Fig. R7).

[Figure]

**Figure R7.** Global map of forest management types in LUH2v2 and irrigation area ratio of cropland for minimum MODIS cropland > 50% in 2011-2020.

We have also changed the sentence in Lines 261-263 now to:

"*We compared the crop irrigation ratio between bins < 10%, 10% to 50% and ≥ 50% for pixels with >50% cropland fraction, which is dominated by cropland in India and North America (Figure 1).*"
to:

*We compared the crop irrigation ratio between bins < 10%, 10% to 50% and ≥ 50% for pixels with >50% cropland fraction, which is dominated by cropland in India (Figure 1).*

**Ln 228-229: "L-VOD is positive in Amazon, central Africa and Southeast Asia regions", difficult to see central Africa and Southeast Asia are positive, are they significant?**

We agree with the reviewer that in central Africa, the pattern is not clear. In the Southeast Asia region, L-VOD showed that pixels with positive drought resistance cover 53.7% of the total vegetated area (MODIS vegetation cover > 0.05) with a significantly positive mean from Student's t-test, as shown in Fig. 3b. We have rephrased the sentence in Line 271:

*…, where drought resistance based on  L-VOD is positive in Amazon and Southeast Asia regions, …*

**Ln 236: clearer compared to what? I can see clearer patterns than those from alpha, but for beta, L-VOD is not as clear as EVI or kNDVI.**

We apologize for the unclear statement. Here the patterns of beta are clearer compared to alpha for all three indices. We have rephrased the sentence in Line 279 to:

*The temperature sensitivity shows a clearer spatial pattern than the resistance to drought duration.*

**Ln 247: Figure 2, double check the unit of temperature coefficients, if it is needed for x axis for (i) and (p)?**

We have removed the wrong unit labels of temperature coefficients in Fig. 2i, 2p (now Fig. A6i, A6p).

**Ln 264: what does the black text in Fig. 3a represent, no regression is applied, or zero coefficients?**

The black text represents that the mean of the distribution of $\alpha$ is not significantly different from zero. We have restructured the caption in Lines 310-312 for clarity:

*... Blue represents significant positive mean of $\alpha$ and red represents significant negative mean of resistance, black indicates regions whose means of the distribution are not significantly different from zero (P-value < 0.05); ...*

In addition, we made the following changes in our main text to make it more precise:
Line 19: changed "L-band mission" to "L-band microwave mission".

Line 19-21: rephrased "L-VOD has the advantage over more commonly used vegetation indices (such as kNDVI, EVI) in that it provides more information on vegetation structure and biomass and suffers from less saturation over dense forests compared." to:
"*Due to its longer wavelength, L-VOD has the advantage over more commonly used vegetation indices (such as kNDVI, EVI) in that it provides different information on vegetation structure and biomass and suffers from less saturation over dense forests.*".

Line 24-25: rephrased "We compare estimates of ecosystem resistance to drought and heat between L-VOD, kNDVI and EVI." to:
"*We compare estimates of ecosystem resistance to drought and heat as retrieved from L-VOD, kNDVI and EVI products.*".

Line 40 and Line 129: CO2 to $CO_2$.

Lines 74-75: changed "... cannot detect biomass changes" to:

"*... cannot detect above-ground biomass changes within the vegetational volume, which represents the amount of carbon stored in above-ground vegetation*".

Lines 84-86: rephrased the sentence "Biomass changes, on the other hand, can be retrieved globally from satellite data, being therefore useful to quantify the ecosystem resistance worldwide." to:

*Above-ground vegetation biomass changes, on the other hand, can be retrieved globally from satellite data, being therefore useful to quantify the ecosystem resistance worldwide (Araza et al., 2023).*

Lines 104-106: rephrased "... we use the L-band passive microwave observations of Vegetation Optical Depth (L-VOD) from the Soil Moisture and Ocean Salinity (SMOS) satellite, …" to:

*…, we use the Vegetation Optical Depth (L-VOD) product based on L-band microwave emission observations from ESA's Soil Moisture and Ocean Salinity (SMOS) satellite, …*

Line 107: changed "As a comparison, we also use EVI and kNDVI." to:

*For comparison, we use EVI and kNDVI.*

Line 114: changed "the SMOS low-frequency microwave satellite" to

"*ESA's SMOS low-frequency passive microwave sensor*"

Line 120: changed "biomass" to "*above-ground biomass*"

Line 121-123: rephrased the sentence "Compared to traditional vegetation indices (NDVI, EVI) and Ku-VOD, X-VOD, C-VOD, L band VOD (L-VOD) is less sensitive to saturation effects at high biomass densities and considers the entire canopy, not just the top layer as for high-frequency VOD" to:

*Compared to traditional vegetation indices (NDVI, EVI) and Ku-, X- or C-VOD, L-band VOD (L-VOD) is less sensitive to saturation effects at high biomass densities and originates from the entire canopy, not just the top layer as for higher frequency VODs.*

Line 125: changed reference "Yang et al., in review, Nat. Geo" in Table 1 to "Yang et al., in press, Nat. Geo"

Line 127-128: changed "The product of L-VOD for the period of 2010–2021 afterwards was filtered strictly (Yang et al., in review), in order to remove the effects of radio frequency interference (RFI) on signals in some regions of the northern hemisphere." to:

*The multi-year L-VOD products were filtered strictly (Yang et al., in press) in order to remove effects of radio frequency interference (RFI) in some regions of the northern hemisphere.*

Line 128: add "then" before "reconstructed"

Line 134: changed "former" to "first"

Line 139: removed "also" in the sentence "L-VOD is also sensitive to …"

Line 147: changed "Moderate Resolution Imaging Spectroradiometer (MODIS)" to "*NASA's Moderate Resolution Imaging Spectroradiometer (MODIS)*"

Line 175: removed "classification" in the sentence "We derived land cover classification based on …"

We have changed the term 'biomes' to 'vegetation categories' because cropland is not a biome in some biome classifications in Lines 179-180:

*We further aggregated the original classes into four vegetation categories (forests, shrublands, grasslands and croplands) according to Table 3.*

and in Lines 241-242:
*We compared the distributions of these groups and distinguished the effect of increasing coverage of specific vegetation categories or some specific land management.*

and in our reply to Ln 150 and in Table 3, which we added in the reply to Ln 162.

Line 226: rephrased "the vegetation L-VOD and greenness" to:
"*the vegetation L-VOD, as an indicator for above-ground vegetation biomass, and EVI, kNDVI as indicators for greenness*"

Line 241: changed "vegetation biomass" to "above-ground vegetation biomass"

Line 294-295: changed "We summarize the results for each of the IPCC AR6 sub-regions in Figure 3. " to:
"*We summarize the results for each of the IPCC AR6 land sub-regions in Figure 3. The full name of each land sub-region is shown in Table A1.*"

Line 338: changed "NDVI" to "kNDVI"

Lines 434-435: changed "... which can be interpreted as an increase in photosynthesis in response to warming" to:
*… which can be interpreted as an increase in photosynthesis in response to warming when enough water is available …*

Line 442-443: changed "the primary forest" to "primary forests" and "secondary forest" to "secondary forests".

Line 477-487: rephrased "L-VOD was produced using low-frequency microwave observations. It has superior sensitivity to carbon density than NDVI, EVI and other higher-frequency VOD products with the ability to penetrate into vegetation with multi-layer canopy and upper part of stem biomass (Rodríguez-Fernández et al., 2018; Tian et al., 2018; Fan et al., 2019; Wigneron et al., 2020). As a result, it is able to retrieve the overall biomass in dense tropical ecosystems when EVI, NDVI and high-frequency L-VOD saturate (Liu et al., 2015). kNDVI overcomes the greenness saturation with increased forest cover empirically but still cannot detect the biomass change under the top canopy layer and other than leaf biomass. In dense forests, L-VOD is also sensitive to woody biomass but EVI and kNDVI only detect the top-layer canopy greenness dynamics. The correlation between L-VOD and kNDVI is also much lower in forests than in cropland and grassland (Figure 6). This is also confirmed in the difference between L-VOD and EVI, kNDVI in dense forests under the tropical climate (Figure A4a-c), where forest canopy structure is generally more complex with higher mean canopy." to:

*ESA's SMOS L-VOD product is calculated from low-frequency, large-wavelength microwave emissions. It has superior sensitivity to carbon density than NDVI, EVI and other higher-frequency VOD products. L-VOD signals originate from deeper volumes of a multi-layer canopy and thus correspond better to the above-ground biomass (Rodríguez-Fernández et al., 2018; Tian et al., 2018; Fan et al., 2019; Wigneron et al., 2020). As a result, it is better capable of retrieving the overall biomass in dense tropical ecosystems, when EVI, NDVI and high-frequency L-VOD saturate (Liu et al., 2015). kNDVI overcomes the greenness saturation with increased forest cover empirically, but still does not detect biomass change under a top canopy layer other than leaf biomass. Generally in dense forests, L-VOD is sensitive to woody biomass where EVI and kNDVI only detect op-layer canopy greenness dynamics. Therefore, the correlation between L-VOD and kNDVI is much lower in forests than in cropland and grassland (Figure 6). This is confirmed in this study in the difference between L-VOD, EVI and kNDVI in dense forests under the tropical climate (Figure A4a-c), where canopy structure is generally more complex and taller.*

Line 535: changed "Supplementary Figures" to "Supplementary Figures and Tables" and added Table A1 to Line 566:

**Table A1: IPCC AR6 reference sub-regions of land**

| Abbreviations | Full name | Abbreviations | Full name |
| --- | --- | --- | --- |
| GIC | Greenland/Iceland | SEAF | S.Eastern-Africa |
| NWN | N.W.North-America | WSAF | W.Southern-Africa |

| | | | |
|---|---|---|---|
| NEN | N.E.North-America | ESAF | E.Southern-Africa |
| WNA | W.North-America | MDG | Madagascar |
| CNA | C.North-America | RAR | Russian-Arctic |
| ENA | E.North-America | WSB | W.Siberia |
| NCA | N.Central-America | ESB | E.Siberia |
| SCA | S.Central-America | RFE | Russian-Far-East |
| NWS | N.W.South-America | WCA | W.C.Asia |
| NSA | N.South-America | ECA | E.C.Asia |
| NES | N.E.South-America | TIB | Tibetan-Plateau |
| SAM | South-American-Monsoon | EAS | E.Asia |
| SWS | S.W.South-America | ARP | Arabian-Peninsula |
| SES | S.E.South-America | SAS | S.Asia |
| SSA | S.South-America | NAU | N.Australia |
| NEU | N.Europe | CAU | C.Australia |
| WCE | West&Central-Europe | EAU | E.Australia |
| EEU | E.Europe | SAU | S.Australia |
| SAH | Sahara | NZ | New-Zealand |
| WAF | Western-Africa | EAN | E.Antarctica |
| CAF | Central-Africa | WAN | W.Antarctica |
| NEAF | N.Eastern-Africa | | |

In Reference: add references for Schwalm et al., 2017; Wu et al., 2022; Yao et al., 2023; Giardina et al., 2018; Araza et al., 2023; De Keersmaecker et al., 2016; Barron-Gafford et al., 2014; Cranko Page et al., 2022; Liu et al., 2019.

**Reference**

Anderegg, W. R. L., Schwalm, C., Biondi, F., Camarero, J. J., Koch, G., Litvak, M., Ogle, K., Shaw, J. D., Shevliakova, E., Williams, A. P., Wolf, A., Ziaco, E., and Pacala, S.: Pervasive drought legacies in forest ecosystems and their implications for carbon cycle models, Science, 349, 528–532, https://doi.org/10.1126/science.aab1833, 2015.

Anderegg, W. R. L., Konings, A. G., Trugman, A. T., Yu, K., Bowling, D. R., Gabbitas, R., Karp, D. S., Pacala, S., Sperry, J. S., Sulman, B. N., and Zenes, N.: Hydraulic diversity of forests regulates ecosystem resilience during drought, Nature, 561, 538–541, https://doi.org/10.1038/s41586-018-0539-7, 2018.

Araza, A., Herold, M., de Bruin, S., Ciais, P., Gibbs, D. A., Harris, N., Santoro, M., Wigneron, J.-P., Yang, H., Málaga, N., Nesha, K., Rodriguez-Veiga, P., Brovkina, O., Brown, H. C. A., Chanev, M., Dimitrov, Z., Filchev, L., Fridman, J., García, M., Gikov, A., Govaere, L., Dimitrov, P., Moradi, F., Muelbert, A. E., Novotný, J., Pugh, T. A. M., Schelhaas, M.-J., Schepaschenko, D., Stereńczak, K., and Hein, L.: Past decade above-ground biomass change comparisons from four multi-temporal global maps, Int J Appl Earth Obs, 118, 103274, https://doi.org/10.1016/j.jag.2023.103274, 2023.

Barron-Gafford, G. A., Cable, J. M., Bentley, L. P., Scott, R. L., Huxman, T. E., Jenerette, G. D., and Ogle, K.: Quantifying the timescales over which exogenous and endogenous conditions affect soil respiration, New Phytol, 202, 442–454, https://doi.org/10.1111/nph.12675, 2014.

Bastos, A., Ciais, P., Friedlingstein, P., Sitch, S., Pongratz, J., Fan, L., Wigneron, J. P., Weber, U., Reichstein, M., Fu, Z., Anthoni, P., Arneth, A., Haverd, V., Jain, A. K., Joetzjer, E., Knauer, J., Lienert, S., Loughran, T., McGuire, P. C., Tian, H., Viovy, N., and Zaehle, S.: Direct and seasonal legacy effects of the 2018 heat wave and drought on European ecosystem productivity, Science Advances, 6, eaba2724, https://doi.org/10.1126/sciadv.aba2724, 2020a.

Bastos, A., Fu, Z., Ciais, P., Friedlingstein, P., Sitch, S., Pongratz, J., Weber, U., Reichstein, M., Anthoni, P., Arneth, A., Haverd, V., Jain, A., Joetzjer, E., Knauer, J., Lienert, S., Loughran, T., McGuire, P. C., Obermeier, W., Padrón, R. S., Shi, H., Tian, H., Viovy, N., and Zaehle, S.: Impacts of extreme summers on European ecosystems: a comparative analysis of 2003, 2010 and 2018, Phil. Trans. R. Soc. B, 375, 20190507, https://doi.org/10.1098/rstb.2019.0507, 2020b.

Briant, G., Gond, V., and Laurance, S. G. W.: Habitat fragmentation and the desiccation of forest canopies: A case study from eastern Amazonia, Biol Conserv, 143, 2763–2769, https://doi.org/10.1016/j.biocon.2010.07.024, 2010.

Broadbent, E. N., Asner, G. P., Keller, M., Knapp, D. E., Oliveira, P. J. C., and Silva, J. N.: Forest fragmentation and edge effects from deforestation and selective logging in the Brazilian Amazon, Biol Conserv, 141, 1745–1757, https://doi.org/10.1016/j.biocon.2008.04.024, 2008.

Buermann, W., Forkel, M., O'Sullivan, M., Sitch, S., Friedlingstein, P., Haverd, V., Jain, A. K., Kato, E., Kautz, M., Lienert, S., Lombardozzi, D., Nabel, J. E. M. S., Tian, H., Wiltshire, A. J., Zhu, D., Smith, W. K., and Richardson, A. D.: Widespread seasonal compensation effects of spring warming on northern plant productivity, Nature, 562, 110–114, https://doi.org/10.1038/s41586-018-0555-7, 2018.

Cranko Page, J., De Kauwe, M. G., Abramowitz, G., Cleverly, J., Hinko-Najera, N., Hovenden, M. J., Liu, Y., Pitman, A. J., and Ogle, K.: Examining the role of environmental memory in the predictability of carbon and water fluxes across Australian ecosystems, Biogeosciences, 19, 1913–1932, https://doi.org/10.5194/bg-19-1913-2022, 2022.

De Keersmaecker, W., van Rooijen, N., Lhermitte, S., Tits, L., Schaminée, J., Coppin, P., Honnay, O., and Somers, B.: Species-rich semi-natural grasslands have a higher resistance but a lower resilience than intensively

managed agricultural grasslands in response to climate anomalies, Journal of Applied Ecology, 53, 430–439, https://doi.org/10.1111/1365-2664.12595, 2016.

Elias, F., Ferreira, J., Lennox, G. D., Berenguer, E., Ferreira, S., Schwartz, G., Melo, L. de O., Reis Júnior, D. N., Nascimento, R. O., Ferreira, F. N., Espirito-Santo, F., Smith, C. C., and Barlow, J.: Assessing the growth and climate sensitivity of secondary forests in highly deforested Amazonian landscapes, Ecology, 101, e02954, https://doi.org/10.1002/ecy.2954, 2020.

Gessler, A., Bottero, A., Marshall, J., and Arend, M.: The way back: recovery of trees from drought and its implication for acclimation, New Phytol, 228, 1704–1709, https://doi.org/10.1111/nph.16703, 2020.

Giardina, F., Konings, A. G., Kennedy, D., Alemohammad, S. H., Oliveira, R. S., Uriarte, M., and Gentine, P.: Tall Amazonian forests are less sensitive to precipitation variability, Nat Geosci, 11, 405–409, https://doi.org/10.1038/s41561-018-0133-5, 2018.

Ingrisch, J. and Bahn, M.: Towards a Comparable Quantification of Resilience, Trends in Ecology & Evolution, 33, 251–259, https://doi.org/10.1016/j.tree.2018.01.013, 2018.

Konings, A. G., Rao, K., and Steele-Dunne, S. C.: Macro to micro: microwave remote sensing of plant water content for physiology and ecology, New Phytologist, 223, 1166–1172, https://doi.org/10.1111/nph.15808, 2019.

Li, X., Wigneron, J.-P., Frappart, F., Fan, L., Ciais, P., Fensholt, R., Entekhabi, D., Brandt, M., Konings, A. G., Liu, X., Wang, M., Al-Yaari, A., and Moisy, C.: Global-scale assessment and inter-comparison of recently developed/reprocessed microwave satellite vegetation optical depth products, Remote Sensing of Environment, 253, 112208, https://doi.org/10.1016/j.rse.2020.112208, 2021.

Liu, L., Chen, X., Ciais, P., Yuan, W., Maignan, F., Wu, J., Piao, S., Wang, Y.-P., Wigneron, J.-P., Fan, L., Gentine, P., Yang, X., Gong, F., Liu, H., Wang, C., Tang, X., Yang, H., Ye, Q., He, B., Shang, J., and Su, Y.: Tropical tall forests are more sensitive and vulnerable to drought than short forests, Global Change Biology, n/a, https://doi.org/10.1111/gcb.16017, 2021.

Liu, Y., Schwalm, C. R., Samuels-Crow, K. E., and Ogle, K.: Ecological memory of daily carbon exchange across the globe and its importance in drylands, Ecol Lett, 22, 1806–1816, https://doi.org/10.1111/ele.13363, 2019.

Qin, Y., Xiao, X., Wigneron, J.-P., Ciais, P., Brandt, M., Fan, L., Li, X., Crowell, S., Wu, X., Doughty, R., Zhang, Y., Liu, F., Sitch, S., and Moore, B.: Carbon loss from forest degradation exceeds that from deforestation in the Brazilian Amazon, Nat. Clim. Chang., 11, 442–448, https://doi.org/10.1038/s41558-021-01026-5, 2021.

Roux, R. L., Wagner, F., Blanc, L., Betbeder, J., Gond, V., Dessard, H., Funatzu, B., Bourgoin, C., Cornu, G., Herault, B., Montfort, F., Sist, P., Begue, A., Dubreuil, V., Laurent, F., Messner, F., Hasan, A. F., and Arvor, D.: How wildfires increase sensitivity of Amazon forests to droughts, Environ Res Lett, 17, 044031, https://doi.org/10.1088/1748-9326/ac5b3d, 2022.

Schwalm, C. R., Anderegg, W. R. L., Michalak, A. M., Fisher, J. B., Biondi, F., Koch, G., Litvak, M., Ogle, K., Shaw, J. D., Wolf, A., Huntzinger, D. N., Schaefer, K., Cook, R., Wei, Y., Fang, Y., Hayes, D., Huang, M., Jain, A., and Tian, H.: Global patterns of drought recovery, Nature, 548, 202–205, https://doi.org/10.1038/nature23021, 2017.

Walther, S., Duveiller, G., Jung, M., Guanter, L., Cescatti, A., and Camps-Valls, G.: Satellite Observations of the Contrasting Response of Trees and Grasses to Variations in Water Availability, Geophysical Research Letters, 46, 1429–1440, https://doi.org/10.1029/2018GL080535, 2019.

Wigneron, J.-P., Fan, L., Ciais, P., Bastos, A., Brandt, M., Chave, J., Saatchi, S., Baccini, A., and Fensholt, R.: Tropical forests did not recover from the strong 2015–2016 El Niño event, Science Advances, 6, eaay4603, https://doi.org/10.1126/sciadv.aay4603, 2020.

Wolf, J., Asch, J., Tian, F., Georgiou, K., and Ahlström, A.: Canopy responses of Swedish primary and secondary forests to the 2018 drought, Environ Res Lett, 18, 064044, https://doi.org/10.1088/1748-9326/acd6a8, 2023.

Wu, C., Peng, J., Ciais, P., Peñuelas, J., Wang, H., Beguería, S., Andrew Black, T., Jassal, R. S., Zhang, X., Yuan, W., Liang, E., Wang, X., Hua, H., Liu, R., Ju, W., Fu, Y. H., and Ge, Q.: Increased drought effects on the phenology of autumn leaf senescence, Nat Clim Change, 12, 943–949, https://doi.org/10.1038/s41558-022-01464-9, 2022.

Yao, Y., Liu, Y., Zhou, S., Song, J., and Fu, B.: Soil moisture determines the recovery time of ecosystems from drought, Global Change Biol, 29, 3562–3574, https://doi.org/10.1111/gcb.16620, 2023.

Zamora-Pereira, J. C., Yousefpour, R., Cailleret, M., Bugmann, H., and Hanewinkel, M.: Magnitude and timing of density reduction are key for the resilience to severe drought in conifer-broadleaf mixed forests in Central Europe, Ann Forest Sci, 78, 68, https://doi.org/10.1007/s13595-021-01085-w, 2021.

**Land-cover and management modulation of ecosystem resistance to drought stress**

Xiao, C., Zaehle, S., Yang, H., Wigneron, J.-P., Schmullius, C., and Bastos, A. *Earth Syst. Dynam. Discuss*

**Response to Reviewer #2**

**R2C1: Xiao et al. make use of several satellite-based products to investigate the ecosystem's resistance to drought stress under different contexts, e.g., forest/cropland, natural/human management. This is an interesting and quite important topic in a warmer world with more droughts.**

We thank the reviewer for the evaluation of our manuscript and for helping to clarify the methods. We have now improved and clarified the methods, following the comments by the reviewer. Please find our point-by-point reply below.

**R2C2: However, I do see some technical flaws that should be improved. First, I'm not convinced by the definition of ecosystem resistance to drought. Why did the authors choose the annual maximum L-VOD/kNDVI/EVI to derive the drought response of the ecosystem? The "time window" when maximum ecosystem productivity occurs is likely in a favorable environmental condition, for instance, very wet, high radiation, and irrigation. I do not think it is a suitable "time window" to "see" drought impacts.**

We thank the reviewer for pointing out the 'time window' to detect drought impacts. First, we would like to point out that we are focusing on biomass and not productivity. In our study, we focused on the inter-annual variations in vegetation state and its responses to drought stress, which can be assessed by annual maximum L-VOD data. In that sense, the maximum value of biomass is more likely to reflect the growing-season integrated productivity, rather than temporary spikes in productivity induced by favorable climate conditions. On the other hand, we agree that such spikes would be reflected in EVI and kNDVI if we were to consider their annual maximum values. To avoid such effects, we used annual mean EVI and kNDVI instead. See more details in our response to R1C4 (i).

There is, however, a more fundamental reason why we chose to work with maximum L-VOD.

The rationale behind the choice of annual maximum L-VOD is that L-VOD is theoretically a closer proxy for vegetation water content (VWC) than aboveground biomass (BM) (Ulaby & Long, 2014). VWC scales with the aboveground vegetation biomass:

$$VOD \sim VWC$$
$$VWC = RWC \times BM$$

where RWC is the relative water content defined as the amount of vegetation water per unit wet biomass (Konings et al., 2019). Therefore, L-VOD is also influenced by short-term variations in RWC and sub-annual L-VOD does not necessarily reflect biomass variations. This is especially problematic during drought periods because part of the reduction of L-VOD results from the short-term reduction in RWC in response to the soil water deficit.

To limit the influence of such short-term variations linked to RWC variability, we used the yearly maximum L-VOD from the reconstructed monthly L-VOD data. The yearly maximum L-VOD, corresponding to the maximum vegetation water content and which is generally associated with wet conditions, is more likely to be decoupled from RWC variations as RWC in the wet season is about constant from year to year (Qin et al., 2021). So the yearly maximum L-VOD is more likely to reflect the changes in aboveground biomass compared to the yearly mean L-VOD.

By doing so, we cannot analyze the immediate effect of drought on biomass, but this is not our objective, as our purpose is to study the concurrent effects of droughts on biomass at annual scales.

**R2C3: Second, in equation (2), "$\varphi$" can represent vegetation memory, which has previously been proposed to be associated with long-term resilience to any type of perturbation. Drought is indeed one of the most important and frequent perturbations. How do we know "$\varphi$" do not already include most drought response information? Do you ever compare the general pattern (not the size due to unit difference) of "$\alpha$" and "$\varphi$"?**

We agree with the reviewer that the $\varphi$ term is closely associated with the vegetation memory and long-term resilience to perturbations (De Keersmaecker et al., 2016) and might reflect intrinsic ecosystem feedbacks such as biomass accumulation or loss (Barron-Gafford et al. 2014). It also reflects the effects of other environmental and climate drivers or nonlinear responses not explicitly considered in our model (Liu et al., 2019). Drought might have legacy effects, which means that droughts that happened in the previous year can continue to influence tree growth (Anderegg et al. 2015). Such effects are partly considered in the $\varphi$ term. However, in our definition of drought resistance, we focused on the concurrent drought effects, but not effects from drought events occurring in the previous year. Following the reviewer's suggestion, we compared the spatial pattern of $\alpha$ and $\varphi$ (Fig. R8). The spatial patterns are not similar and spatial correlations are close to 0 from L-VOD (0.008), EVI (-0.008), and kNDVI (0.011), so drought resistance is unlikely to be influenced by vegetation memory term.

We have tested whether including the $\varphi$ term would affect the $\alpha, \beta$ coefficients and contribute to increased adjusted $R^2$, see more details in our responses to R1C4 (iii).

[Figure]

**Figure R8.** Spatial distribution of ecosystem resistance to drought duration $\alpha$ (a, b, c) and the vegetation memory sensitivity $\varphi$ (e, f, g) in the linear autoregressive model with memory term from L-VOD, EVI and kNDVI. The averages for different latitudes and their standard deviations are shown on the right (d, h).

**R2C4: Third, $N$ is the centered number of drought months in each year. A bit confused, what does "centered" mean?**

We meant that the number of drought months in each year has been subtracted from their average for 2010-2020, which is shown in the equation: $X_{year} - \overline{X}$. This does not influence the results in our linear regression except for the model intercept. Therefore we have removed this operation from our analysis.
We have modified the description in Line 208:

*N is the number of drought months in each year in 2010-2020.*

**R2C5: Fourth, you say "To calculate the ecosystem resistance to heat and drought" in Line 182, but you utilize the yearly mean 2m temperature as the proxy of "heat".**

We agree with the review that the yearly mean 2m temperature should not be considered as the proxy for heat. We have modified our definition of resistance and rephrased it to 'temperature sensitivity'. We included this term to account for the fact that vegetation growth is also strongly controlled by temperature. See more details in our reply to R1C4 (ii).

**R2C6: In summary, this is an important and interesting paper. I suggest the authors refine the method. I'm happy to review this paper again.**

We thank the reviewer for the constructive comments on our methods and we have improved the above points in our revised manuscript.

**Reference**

Anderegg, W. R. L., Schwalm, C., Biondi, F., Camarero, J. J., Koch, G., Litvak, M., Ogle, K., Shaw, J. D., Shevliakova, E., Williams, A. P., Wolf, A., Ziaco, E., and Pacala, S.: Pervasive drought legacies in forest ecosystems and their implications for carbon cycle models, Science, 349, 528–532, https://doi.org/10.1126/science.aab1833, 2015.

Barron-Gafford, G. A., Cable, J. M., Bentley, L. P., Scott, R. L., Huxman, T. E., Jenerette, G. D., and Ogle, K.: Quantifying the timescales over which exogenous and endogenous conditions affect soil respiration, New Phytol, 202, 442–454, https://doi.org/10.1111/nph.12675, 2014.

De Keersmaecker, W., van Rooijen, N., Lhermitte, S., Tits, L., Schaminée, J., Coppin, P., Honnay, O., and Somers, B.: Species-rich semi-natural grasslands have a higher resistance but a lower resilience than intensively managed agricultural grasslands in response to climate anomalies, Journal of Applied Ecology, 53, 430–439, https://doi.org/10.1111/1365-2664.12595, 2016.

Konings, A. G., Holtzman, N. M., Rao, K., Xu, L., and Saatchi, S. S.: Interannual Variations of Vegetation Optical Depth are Due to Both Water Stress and Biomass Changes, Geophysical Research Letters, 48, e2021GL095267, https://doi.org/10.1029/2021GL095267, 2021.

Liu, Y., Schwalm, C. R., Samuels-Crow, K. E., and Ogle, K.: Ecological memory of daily carbon exchange across the globe and its importance in drylands, Ecol Lett, 22, 1806–1816, https://doi.org/10.1111/ele.13363, 2019.

Qin, Y., Xiao, X., Wigneron, J.-P., Ciais, P., Brandt, M., Fan, L., Li, X., Crowell, S., Wu, X., Doughty, R., Zhang, Y., Liu, F., Sitch, S., and Moore, B.: Carbon loss from forest degradation exceeds that from deforestation in the Brazilian Amazon, Nat. Clim. Chang., 11, 442–448, https://doi.org/10.1038/s41558-021-01026-5, 2021.

Ulaby, F. T., & Long, D. G.: Microwave radar and radiometric remote sensing. Ann Arbor: The Univerity of Michigan Press, 2014

---

## Author Response (AR2)

**Land cover and management effects of ecosystem resistance to drought stress**

Xiao, C., Zaehle, S., Yang, H., Wigneron, J.-P., Schmullius, C., and Bastos, A. *Earth Syst. Dynam. Discuss*

**Response to Reviewer #1**

**R1C1: Thank you to the authors for your comprehensive responses and the additional analyses, which have successfully addressed many of my initial concerns. I have a few remaining thoughts on aspects of the manuscript that could be clarified further.**

Thank you for the constructive comments and detailed suggestions.

**R1C2: I was particularly impressed with the findings presented in Figure 5. Your large-scale remote-sensing based analysis has yielded some really intriguing results. One point that still wasn't entirely clear to me, however, is your use of the term "similar background climate." I understand that you employed a multiple linear regression model to control for temperature and precipitation, but missing the consideration of climate seasonality when using the annual mean. Even regions with the same mean climate can differ significantly for the coldest and warmest months. In addition, a consideration of radiation variability could strengthen the model, especially for high-latitude regions where both primary and secondary forests are present (important for interpreting Fig. 5a). Your Figure R4 already hints at the weak relationship between T2m and drought duration in boreal regions, suggesting that radiation could be an important factor there.**

(i) Thank you for suggesting taking climate seasonality into consideration when we define a similar climate background. When we define the climate space, there is a trade-off between the number of available pixels for comparison and the details of our climate space. Since we have a limited area with significant drought resistance due to the short time series of L-VOD, the noise in the L-VOD data or areas where droughts do not strongly influence the vegetation growth in 2010-2020, we used only climatological mean temperature and precipitation to define our climate space.

We added the following sentences in Lines 545-550 in Section 4.5:
 … *Even though we controlled for similar climate backgrounds by aggregating pixels based on their long-term temperature and precipitation averages, there are other climate effects that were not considered in our statistical analysis, for example, the interannual variability of precipitation and climate seasonality of temperature. With only limited areas exhibiting significant drought resistance α, and given the need to ensure a large enough*

*number of pixels for comparison in a similar climate space, it remains challenging to disentangle the potential confounding effects of all the climate variables and their variabilities.*

(ii) In the boreal regions, radiation is an important limitation factor for vegetation growth (Seddon et al., 2016). However, the surface incoming shortwave radiation strongly correlates with the yearly total precipitation (Figure R1a), yearly mean temperature (Fig. R1b), and soil moisture drought duration on land (Fig. R1c).

Precipitation is closely related to cloud cover, which directly influences the incoming solar radiation at the land surface. The air temperature at 2 m increases due to a higher energy input when surface incoming solar radiation is higher. Droughts are associated with clear-sky conditions that favor more incoming solar radiation (O et al., 2022).

[Figure]

**Figure R1.** Temporal correlation between yearly mean surface incoming solar radiation and (a) yearly total precipitation (b) yearly mean temperature at 2 m and (c) drought duration (months per year) in 1979-2020.

To avoid the influence of collinearity on vegetation sensitivity to temperature and drought duration, and since we only have ten years of data for the regression with L-VOD, we decided not to incorporate radiation into our linear regression model. Nevertheless, we acknowledge that the trade-off between drought and radiation can partly explain some of the patterns of high drought resistance that we find in energy-limited areas, e.g. in high latitudes.

Therefore, we evaluate how our results change when controlling for radiation by adding surface incoming solar radiation as an additional predictor. Fig. R2 shows that adjusted $R^2$ increased by 0.04 on average from L-VOD. The median of the adjusted $R^2$ difference is -0.03. Adjusted $R^2$ does not increase in most areas except for the boreal eastern Siberia. In the region north of 60° N, the averaged adjusted $R^2$ increased by 0.126. Drought resistance shows an average difference of -0.009 month$^{-1}$ and a median difference of -0.001 month$^{-1}$. Temperature sensitivity shows an average difference of -0.003 and a median difference of 0.001. For EVI and kNDVI, adjusted $R^2$ increased by 0.05 and 0.084 on average. The median of the adjusted $R^2$ difference is -0.02 and 0.003. The adjusted $R^2$ increases in southern South America, southern Australia, central North America, and central Eurasia regions, where

surface incoming solar radiation is highly correlated with temperature at 2 m, which can lead to problems in interpreting the coefficient of temperature. In the region north of 60° N, the averaged adjusted $R^2$ only increased by 0.012 and 0.014.

[Figure]

**Figure R2.** Difference between linear regression model with/without surface incoming solar radiation as an additional predictor for ecosystem resistance to drought duration $\alpha$ from L-VOD, EVI, and kNDVI (a, b, c). Similar for temperature sensitivity $\beta$ (e, f, g) and adjusted $R^2$ (i, j, k). The averages for different latitudes and their standard deviations are shown on the right (d, h, l).

We then compare (Fig. R3) how the values of $\alpha$ and $\beta$ change. For significant drought resistance and temperature sensitivity coefficients (*P*-value < 0.05), which we have used for further statistical analysis of land cover and management effects, the results are similar to regression without radiation or with radiation in predictors. The correlation between $\alpha$ (without radiation) and $\alpha$ (with radiation) is close to 1 (Fig. R3a-c). Similar results are found for $\beta$ (Fig. R3d-f).

[Figure]

**Figure R3.** Difference between estimated significant drought resistance $\alpha$ (a-c) and temperature sensitivity $\beta$ (d-f) (*P*-value < 0.05) with/without surface incoming solar radiation as an additional predictor.

We further evaluate how the effect of land cover on drought resistance changes if we consider radiation as an additional predictor for the pixels with increased adjusted $R^2$. Fig. R4 shows that the similar significant $\alpha$ and $\beta$ do not change our main conclusion. Dominant forests are more resistant than dominant crops to drought stress. Although there are still some changes in the significance test between groups: the significance of the contrast between $\alpha$ in >75% forests and >75% crops calculated from EVI becomes non-significant.

[Figure]

**Figure R4.** Ecosystem resistance to drought and temperature after including surface incoming solar radiation as an additional predictor only for areas where adjusted $R^2$ increases, binned for different levels of the aggregated forest and cropland fraction classes from the three land cover products (a) L-VOD, (b) EVI and (c) kNDVI for drought resistance coefficients $\alpha$ and (d-f) for temperature resistance coefficients $\beta$. Only significant coefficients $\alpha$ in the linear model ($P$-value < 0.05) are included and groups with less than 20 pixels are excluded. The number in each bin is the number of pixels in this category. Only pixels with no change in 25% bins of the four dominant vegetation categories (forests, shrublands, grasslands, and croplands) are analyzed. The star on the upper right corner indicates significantly higher resistance in forest > 75% than crop > 75% at the 0.05 significance level from the unpaired two-sample Wilcoxon test.

Finally, we evaluate whether adding radiation as an additional predictor influences our conclusion that primary forests are more resistant to drought than secondary forests (Fig. R5). Primary forests are still more resistant to drought stress than secondary forests and the difference is most obvious in L-VOD, although there are still some changes in the significance test between groups: a higher resistance $\alpha$ in primary forests than in secondary forests change from non-significant to significant in kNDVI.

[Figure]

**Figure R5.** Similar to Figure 5a in the original manuscript, but including the effects of surface incoming solar radiation only for areas where adjusted $R^2$ increases.

Considering the risk of overfitting for our small samples and the strong correlation between radiation and temperature as well as drought duration, we decided to keep the original predictors, temperature and drought duration to investigate the ecosystem resistance to drought stress.

We added the following explanation of the choice of predictors in Lines 227-232:

*We note that radiation plays an important role in the energy-limited boreal region. However, surface incoming solar radiation strongly correlates with temperature and drought duration. The air temperature at 2 m increases due to a higher energy input when surface incoming solar radiation is higher. Droughts are associated with clear-sky and sunnier conditions that favor more incoming solar radiation (O et al., 2022). To avoid the influence of collinearity on estimated vegetation sensitivity to temperature and drought duration, and given that only ten years of data are available, we did not incorporate radiation into our linear regression model.*

And reference in Lines 895-896:

*O, S., Bastos, A., Reichstein, M., Li, W., Denissen, J., Graefen, H., and Orth, R.: The Role of Climate and Vegetation in Regulating Drought–Heat Extremes, J Climate, 35, 5677–5685, https://doi.org/10.1175/JCLI-D-21-0675.1, 2022.*

We have discussed the potential effect of radiation in boreal regions in Lines 430-432 in our submitted manuscript before:

*For boreal regions, the soil is generally humid, so that drought defined as the 10th percentile of the corresponding distribution likely still provides critical water storage for vegetation. The potential environmental limitations to vegetation growth in these areas are temperature and radiation rather than water availability (Boisvenue and Running, 2006).*

**R1C3: I also appreciated your comparative analysis of alpha based on the growing season versus the whole year. However, I don't agree that "the calculated drought resistance is the same" as you stated in your reply. I am actually a bit surprised by the resulting goodness of fit (now showing an explained 36% variation of the original alpha) when only focusing on parts of the northern hemisphere. I would expect even more deviation could occur when including the southern hemisphere, given the strong effects of elevation on the growing season there. I suggest an explicit discussion of the potential impacts of the choice of aggregation period as the growing season on the analysis. Any insights into this would be fascinating and valuable.**

We apologize for the unclear statement that *'the calculated drought resistance is the same'*. In our reply, we tried to describe different situations when drought resistance was significantly changed when calculating drought duration only for the growing season, instead of the whole year. In this statement, we tried to express that there exist some pixels where the whole year drought duration is exactly the same as the growing drought duration, so that the calculated drought resistance $\alpha$ in those pixels does not change. However, in our reply, we also described other situations when absolute values of $\alpha_{gs}$ become higher or $\alpha_{year}$ is above 1 but $\alpha_{gs}$ becomes close to 0 (See Reply (i) to Comment R1C4 of our previous replies "... *In 75 pixels, the absolute values of $\alpha_{year}$ decrease from values above 1 to values below 0.1 of $\alpha_{gs}$ as shown by the values close to the zero line in Fig. R3...*"). In general, indeed the conclusion is that they are not the same.

We have shown in our reply to the previous R1C4 that there is a good correlation (0.8) between the calculated alpha based on whole-year drought duration ($\alpha_{year}$) and growing-season drought duration ($\alpha_{gs}$) in the Northern Hemisphere extratropics. That is, $\alpha_{gs}$ explains 64% of the variance of $\alpha_{year}$. Only 8.6% of the pixels changed their signs of $\alpha$ between the two calculations. We also applied the same analysis for the globe using the phenology data from MODIS MCD12Q2 and defined the growing season with the long-term mean green-up day and dormancy day of the year. The results are shown in Fig. R6 and Fig. R7 shown below.

At the global scale, $\alpha_{gs}$ shows a spatial correlation of 0.66 with $\alpha_{year}$ (Fig. R8a). The average difference is -0.003 month$^{-1}$. If filtered only to significant $\alpha$ ($P$-value < 0.05), the spatial correlation between $\alpha_{gs}$ and $\alpha_{year}$ is 0.92 (Fig. R8b). Similar to what we found in the Northern Hemisphere extratropics, the spatial contrast of negative values and positive values of $\alpha_{gs}$ and $\alpha_{year}$ are still similar. The value of $\alpha$ switched signs between the two calculations only for 0.1% of those pixels where $\alpha_{gs}$ and $\alpha_{year}$ are both significant. 83.0% of the pixels where both $\alpha_{gs}$ and $\alpha_{year}$ are significant increase their absolute values due to a shorter drought duration when considering only the growing season drought. Temperature sensitivity $\beta_{year}$ and $\beta_{gs}$ show stronger correlation of 0.94 (Fig. R8c-d). Overall, our main results for the differences between forests and crops, and the effects of land management practices still hold. Similar results are found in EVI and kNDVI (not shown for conciseness).

[Figure]

**Figure R6.** Comparison between the linear regression model with the whole year drought or growing season drought duration for ecosystem resistance to drought duration $\alpha_{\text{year}}$ (a) and $\alpha_{\text{gs}}$ (b) from L-VOD. The differences in $\alpha$ and $R^2$ (model with whole year drought minus growing season are shown in (c, d).

[Figure]

**Figure R7.** Comparison between the linear regression model with the whole year drought or growing season drought duration for temperature sensitivity $\beta$ (a, b) from L-VOD. The differences in $\beta$ and $R^2$ (model with whole year drought minus growing season) are shown in (c, d).

[Figure]

**Figure R8.** Comparison between ecosystem resistance to the whole year drought duration and growing season drought duration. (a) For drought resistance $\alpha$; (b) For significant drought resistance $\alpha$ ($P$-value < 0.05); (c) For temperature sensitivity $\beta$; (d) For significant temperature sensitivity $\beta$ ($P$-value < 0.05).

We added the following sentences in Lines 434-436 accordingly:

*We also evaluated whether limiting drought duration to the growing season of each year. The resulting $\alpha$ and $\beta$ values over pixels where the coefficients are significant (P-value < 0.05) are strongly correlated to $\alpha$ and $\beta$ calculated based on whole-year drought duration and results still hold.*

**R1C4: In summary, I find the study to be filled with novel insights and am excited about its contributions. Further clarification is needed for more compelling findings and could make the work even more impactful.**

Thank you again for the insightful comments and suggestions. We added some sentences to our main text in our reply to R1C2 and R1C3 for clarification and we also pasted them here:

We added the following sentences in Lines 545-550 in Section 4.5:

*… Even though we controlled for similar climate backgrounds by aggregating pixels based on their long-term temperature and precipitation averages, there are other climate effects that were not considered in our statistical analysis, for example, the interannual variability of precipitation and climate seasonality of temperature. With only limited areas exhibiting significant drought resistance α, and given the need to ensure a large enough number of pixels for comparison in a similar climate space, it remains challenging to disentangle the potential confounding effects of all the climate variables and their variabilities.*

We added the following explanation of the choice of predictors in Lines 227-232:

*We note that radiation plays an important role in the energy-limited boreal region. However, surface incoming solar radiation strongly correlates with temperature and drought duration. The air temperature at 2 m increases due to a higher energy input when surface incoming solar radiation is higher. Droughts are associated with clear-sky and sunnier conditions that favor more incoming solar radiation (O et al., 2022). To avoid the influence of collinearity on estimated vegetation sensitivity to temperature and drought duration, and given that only ten years of data are available, we did not incorporate radiation into our linear regression model.*

And reference in Lines 895-896:

*O, S., Bastos, A., Reichstein, M., Li, W., Denissen, J., Graefen, H., and Orth, R.: The Role of Climate and Vegetation in Regulating Drought–Heat Extremes, J Climate, 35, 5677–5685, https://doi.org/10.1175/JCLI-D-21-0675.1, 2022.*

We added the following sentences in Lines 434-436 accordingly:

*We also evaluated whether limiting drought duration to the growing season of each year. The resulting α and β values over pixels where the coefficients are significant (P-value < 0.05) are strongly correlated to α and β calculated based on whole-year drought duration and results still hold.*

**Reference**

O, S., Bastos, A., Reichstein, M., Li, W., Denissen, J., Graefen, H., and Orth, R.: The Role of Climate and Vegetation in Regulating Drought–Heat Extremes, J Climate, 35, 5677–5685, https://doi.org/10.1175/JCLI-D-21-0675.1, 2022.

Seddon, A. W. R., Macias-Fauria, M., Long, P. R., Benz, D., and Willis, K. J.: Sensitivity of global terrestrial ecosystems to climate variability, Nature, 531, 229–232, https://doi.org/10.1038/nature16986, 2016.

**Land cover and management effects of ecosystem resistance to drought stress**

Xiao, C., Zaehle, S., Yang, H., Wigneron, J.-P., Schmullius, C., and Bastos, A. *Earth Syst. Dynam. Discuss*

**Response to Reviewer #2**

**R2C1: I thank the authors for their efforts in revision. Most of my concerns are resolved. However, I still have some questions.**
**1) As mentioned by Reviewer 1, do you have estimates of the explanatory power of regression (2) for each pixel? Which region do your regressions explain very well? As I stated in the last round, apart from drought months and temperature, other factors can be important for biomass/productivity interannual variability, especially in croplands with human management. I would suggest the authors only focus on regions where your regressions can work very well or drought months are very important.**
**In addition, the authors at least need to acknowledge that the model might be not able to perfectly disentangle drought and temperature effects given the strong collinearity between them shown in Figure R4.**

Thanks for the suggestions, they helped to clarify the results and better organize the methods. We estimated the explanatory power of regression with $R^2$ and the results are shown in Fig. R9. We also added it in our Appendix A as Fig. A7. Our regression explains well in some areas in eastern South America, southern Africa, eastern Australia, and some boreal regions where the $R^2$ is higher than 0.5. At the global scale, approximately 15%, 23%, and 27% of the pixels exhibit an $R^2$ exceeding 0.5 when derived from L-VOD, EVI, and kNDVI, respectively.

We added the following sentences in Lines 312-314 to describe the spatial distribution of $R^2$:

*Our model performs better in some areas in eastern South America, southern Africa, eastern Australia, and some boreal regions, where the $R^2$ is higher than 0.5 (Figure A7). At the global scale, approximately 15%, 23%, and 27% of the pixels exhibit an $R^2$ exceeding 0.5 when derived from L-VOD, EVI, and kNDVI, respectively.*

[Figure]

**Figure R9.** Spatial map of $R^2$ for (a) L-VOD, (b) EVI, (c) kNDVI. The averages for different latitudes and their standard deviations are shown on the right (d).

We agree with the reviewer that other factors beyond drought duration and temperature can be important. To address this, we consider only pixels where the drought duration or temperature significantly influences the L-VOD, kNDVI, or EVI at a significance level of 5% in subsequent analyses where we investigate the effect of land cover and land management practices (Fig. 3c, 4, 5).

We agree with the reviewer that the model might not be able to perfectly disentangle the drought and temperature effects given the strong collinearity in some tropical regions (Fig. R10). We added Fig. R10 in supplementary as new Fig. A8. and added the following sentences in Lines 552-554 (Section 4.5):

*Drought duration shows a high correlation with yearly mean temperature in some regions in northern Amazon, Southern Africa and South Asia (Figure A8), so the multiple linear regression model might not perfectly disentangle their effects in these areas. We avoid these issues by analyzing those pixels with significant values of α and β (P-value < 0.05).*

[Figure]

**Figure R10.** Temporal correlation between yearly mean temperature and drought duration (months per year) in 1979-2020.

Following the reviewer's remark, we now consider the effects of crop and forest wood harvest (based on LUH2 v2h) on drought resistance. However, the crop harvest fraction provided by

LUH2 v2h is homogeneous over the globe (Fig. R11), so that such information does not aid our analysis.

Forest wood harvest in LUH2 v2h is smaller than 1% of the respective forest area for more than 90% of vegetated pixels (vegetation cover ≥ 5%) (Fig. R12). Therefore, we tested that the effect on our main results is residual (not shown). However, this information allows us to compare drought resistance for different wood harvest intensities in forests, analogous to our analysis of crop irrigation. We found that forest-dominated pixels with more intense harvest activity (wood harvest intensity ≥ 10%) tend to be less resistant to drought than forest-dominated pixels with virtually no harvest (< 1% forests are harvested) (Fig. R13).

[Figure]

**Figure R11.** Averaged fraction of crop harvest annually for (a) C3 perennial crops, (b) C4 perennial crops in 2010-2020.

[Figure]

**Figure R12.** Averaged wood harvest fraction of forest area annually in 2010-2020.

[Figure]

**Figure R13.** Ecosystem resistance to droughts for different forest wood harvest area ratio of forest area. Only significant coefficients $\alpha$ in the linear model (*P*-value < 0.05) are included. Stars indicate that the median value of a given category is greater than the median of the previous category and triangles indicate the median of this category is greater than the median of the first category at the 0.05 significance level from the unpaired two-sample Wilcoxon test. The number in each box is the number of bins/pixels in this category. Only pixels with > 50% forest cover in 2011-2020 were selected.

We added the analysis method in Lines 190-192:

*… It also provides wood harvest area as a fraction of the total grid cell area. We converted this to the fraction of wood harvest from forests (described below as forest wood harvest intensity) by dividing the wood harvest area by the forest area fraction of the total grid cell area. We limit this analysis to pixels with > 50% forest cover.*

We changed the "We finally investigated the ecosystem resistance $\alpha$ for different irrigation levels (Figure 5c)." in Line 308 to:

*We also investigated the ecosystem resistance $\alpha$ for different irrigation levels (Figure 5c).*

We added Fig.R13 as Fig.5d in our manuscript and the following sentences in results in Lines 404-407:

*We finally explored the potential role of forest wood harvest intensity (Figure 5d). All three satellite products agree on a significant decrease of drought resistance ($\alpha$) with increased forest wood harvest intensity, from a median of -0.21 month$^{-1}$ under < 1% harvest area ratio, to -0.34 month$^{-1}$ under 1-10% wood harvest intensity, and -0.40 month$^{-1}$ for >10% harvest intensity, based on L-VOD. Results from EVI and kNDVI are consistent with those of L-VOD.*

We also added the following sentence to our discussion section 4.3 in Lines 503-510:

*Our results indicate that forests with higher harvest intensities tend to be less resistant to drought globally. In-situ studies in different biomes show that forest management can influence forest resistance to disturbances such as drought (Silva Junior et al., 2020; Fawcett et al., 2022). This could be linked to the more complex structure of dense forests, whose below canopy microclimate might help to buffer forest stands from macroclimatic temperature extremes, e.g., in temperate broadleaved and mixed forest biome (Sanczuk et al., 2023). Forest thinning, depending on its intensity, has also been reported to result in lower drought resistance and resilience in older mature forests in north temperate forest ecosystems. This might be due to trees reaching larger sizes during stand development, which in turn increases water demand during droughts (D'Amato et al., 2013).*

We also added a sentence in Line 32:
*Forest harvest decreases the drought resistance of forests.*

We added the new references in Lines 745, 773, 929, and 950:

*D'Amato, A. W., Bradford, J. B., Fraver, S., and Palik, B. J.: Effects of thinning on drought vulnerability and climate response in north temperate forest ecosystems, Ecol Appl, 23, 1735–1742, https://doi.org/10.1890/13-0677.1, 2013.*

*Fawcett, D., Sitch, S., Ciais, P., Wigneron, J. P., Silva-Junior, C. H. L., Heinrich, V., Vancutsem, C., Achard, F., Bastos, A., Yang, H., Li, X., Albergel, C., Friedlingstein, P., and Aragão, L. E. O. C.: Declining Amazon biomass due to deforestation and subsequent degradation losses exceeding gains, Global Change Biology, n/a, https://doi.org/10.1111/gcb.16513, 2022.*

*Sanczuk, P., De Pauw, K., De Lombaerde, E., Luoto, M., Meeussen, C., Govaert, S., Vanneste, T., Depauw, L., Brunet, J., Cousins, S. A. O., Gasperini, C., Hedwall, P.-O., Iacopetti, G., Lenoir, J., Plue, J., Selvi, F., Spicher, F., Uria-Diez, J., Verheyen, K., Vangansbeke, P., and De Frenne, P.: Microclimate and forest density drive plant population dynamics under climate change, Nat Clim Change, 1–8, https://doi.org/10.1038/s41558-023-01744-y, 2023.*

*Silva Junior, C. H. L., Aragão, L. E. O. C., Anderson, L. O., Fonseca, M. G., Shimabukuro, Y. E., Vancutsem, C., Achard, F., Beuchle, R., Numata, I., Silva, C. A., Maeda, E. E., Longo, M., and Saatchi, S. S.: Persistent collapse of biomass in Amazonian forest edges following deforestation leads to unaccounted carbon losses, Sci Adv, 6, eaaz8360, https://doi.org/10.1126/sciadv.aaz8360, 2020.*

Finally, based on these results, we have decided to analyze the forest age effect on drought resistance only for primary tropical evergreen broadleaf forests (EBF), rather than tropical EBF as in the previous version of the manuscript. This is because we have already shown a significant difference between the primary and secondary forests (reproduced Fig. 5b as Fig. R14 here) and the age structure of secondary forests is expected to be influenced by management practices. The result is similar to the previous, with an increased average in

drought resistance $\alpha$ with increasing forest age. The median of $\alpha$ is significantly higher in forests aged 100-300 years and older than 300 years. Therefore we modified the sentences in Lines 266-269 to:

*..., we selected pixels with dominant tropical evergreen broadleaf forests, >50% forest fraction to avoid confounding effects of management over secondary forests. We then selected only pixels belonging to the primary forests we defined above and grouped the forest ages into three groups [0, 100), [100, 300) and ≥ 300 years.*

We changed previous Lines 390-396 to:

*We further tested the effect of forest ages in modulating the ecosystem resistance in the tropical primary evergreen broadleaf forest. Based on L-VOD, forests older than 100 years are substantially more resistant to drought than forests younger than 100 years. The median of $\alpha$ for forests younger than 100 years is -0.549 month$^{-1}$, while the median values of $\alpha$ for forests aged 100-300 years and older than 300 years are 0.455 month$^{-1}$ and 0.360 month$^{-1}$ respectively. We also find a significant (P-value < 0.05) increase of $\alpha$ in kNDVI between forests aged 100-300 years and older than 300 years, but the effect is not as large as in L-VOD and we found no significant differences based on EVI. These results indicate that VOD is more sensitive to water volume and biomass than reflectance indices in general.*

We changed "In tropical evergreen broadleaf forests" in Line 30 to:
*In tropical primary evergreen broadleaf forests…*

[Figure]

**Figure R14.** Ecosystem resistance to droughts for different forest ages in the tropical primary EBF. Only significant coefficients $\alpha$ in the linear model (*P*-value < 0.05) are included. Stars indicate the median value of this category is greater than the median of the previous category and triangles indicate the median of this category is greater than the median of the first

category at the 0.05 significance level from the unpaired two-sample Wilcoxon test. The number in each box is the number of bins/pixels in this category.

Finally, in the discussion, we refer to other factors that might influence the relationships we find in Lines 556-564:

> *Other factors related to land management, e.g., different crop rotations or harvest intensities, also play an important role in changing the vegetation biomass or greenness, especially in croplands, and might influence drought resistance and temperature sensitivity. The LUH2 v2h dataset provides additional information about crop and wood harvest practices. Crop harvest in LUH2v2 is spatially homogeneous so that it cannot be used to evaluate spatial differences in drought and temperature sensitivity over croplands. Forest wood harvest in LUH2 v2h is smaller than 1% of the respective forest area for more than 90% of vegetated pixels (vegetation cover ≥ 5%). Therefore, we tested that the effect on our main results for primary forests and forest age is residual. For a more detailed analysis of other management practices, higher-resolution data on vegetation and management would be needed.*

**R2C2: 2) Figure 2: The color is not reader-friendly. It looks like very limited areas show significant drought resistance at a 10% significance level? Could you add a statistical analysis on this? Can you mask regions that show insignificant values using the color of white instead?**

We agree with the reviewer that there are only limited areas showing significant drought resistance at a 10% significance level. To constrain the uncertainty, we only used the pixels showing significant drought resistance at a 5% significance level for all the statistical analyses.

Thanks for suggesting a statistical analysis of the significance and uncertainty of drought resistance. We added a more detailed description about the percentage of the pixels showing significant drought resistance at 10% and 5% significance levels in Lines 286-290 (Section 3.1):

> *In our analysis, we observe that 12% of pixels show significant drought resistance at a 10% significance level (6%, 6%, and 7% at a 5% significance level) from L-VOD, EVI, and kNDVI. We only used the significant drought resistance at a 5% significance level to investigate the impacts of land cover and land management, ensuring that the vegetation growth is impacted by drought conditions.*

We also added more descriptions on the percentage of the pixels showing significant temperature sensitivity at 10% and 5% significance levels in Lines 300-303 (Section 3.1):

> *15%, 26%, and 31% of pixels show significant temperature sensitivity at a 10% significance level (9%, 17%, and 21% at a 5% significance level) from L-VOD, EVI, and kNDVI. We only used these pixels to investigate the land cover and land management effects to make sure that the vegetation growth is relevant to temperature.*

Thanks for suggesting a better color scheme to visualize the drought resistance pattern. We also checked whether our color scheme is color-blind friendly. We now modified Figure 2 (reproduced here as Fig. R15) using white color to mask the insignificant values and a new color palette in Figure 2d, h and changed the color range for $\alpha$ from -1-1 to -0.5-0.5. We changed the figure caption in Lines 308-310 to:

*The pixels with non-significant $\alpha$ and $\beta$ at a 10% significance level are masked with white color. The full-page figures where pixels with non-significant $\alpha$ and $\beta$ at a 5% significance level are masked with white color are provided in the supplementary Figure S1-S3 for better visualization.*

[Figure]

**Figure R15.** Ecosystem resistance to drought duration and temperature sensitivity. Spatial map of drought resistance $\alpha$ for (a) L-VOD, (b) EVI, (c) kNDVI. Same for temperature sensitivity $\beta$ (e, f, g). The averages for different latitudes and their standard deviations are shown on the right (d, h). The pixels with non-significant $\alpha$ and $\beta$ at a 10% significance level are masked with white color. The full-page figures where pixels with non-significant $\alpha$ and $\beta$ at a 5% significance level are masked with white color are provided in the supplementary Fig. S1-S3 for better visualization.

We also changed the color palette for Fig. A6 and Fig. B1 and reproduced them here as Fig. R16 and Fig. R17:

[Figure]

**Figure R16.** Ecosystem resistance to drought duration and its standard error. Spatial map of drought coefficients $\alpha$ for (a) L-VOD, (b) EVI, (c) kNDVI and their standard error (e, f, g). Same for temperature coefficients $\beta$ (i, j, k) and their standard error (m, n, o). The averages for different latitudes and their standard deviations are shown on the right (d, h, l, p).

[Figure]

**Figure R17.** Difference between linear regression model with/without memory term (model with $\varphi$ minus model without $\varphi$) for ecosystem resistance to drought duration $\alpha$ from L-VOD, EVI and kNDVI (a, b, c). Similar for temperature sensitivity $\beta$ (e, f, g) and adjusted $R^2$ (i, j, k). The averages for different latitudes and their standard deviations are shown on the right (d, h, l).

**R2C3: 3) Figure 3: Here you used the significance level of 5%. Any reason?**

Thank you for pointing it out. We used a significance level of 5% in Figure 3 and all the subsequent statistical analyses. As discussed in R1C2, there are only limited areas showing significant drought resistance. We applied a less strict significance level of 10% in Figure 2 to better visualize the spatial pattern of drought resistance. However, for all subsequent statistical analyses, we used a significance level of 5% for a more robust analysis of the relevance of land use and management by reducing the fraction of false positives. Nevertheless, we acknowledge that there is an inconsistency between the map in Figure 2 and the subsequent figures. Therefore, we will add full-page figures in the supplement with the pixels selected for subsequent analysis (see at the end of the replies Fig. R18-20 as Fig. S1-3 in our supplementary).

**R2C4: I suggest the authors revise the paper. I'm happy to review it again.**

Thanks again for the constructive comments and suggestions during the review process.

[Figure]

**Figure R18.** Ecosystem resistance to drought duration and temperature sensitivity selected for our analysis of land cover and management effects (*P*-value < 0.05). Spatial map of (a) drought resistance $\alpha$ and (b) temperature sensitivity $\beta$ for L-VOD.

[Figure]

**Figure R19.** Ecosystem resistance to drought duration and temperature sensitivity selected for our analysis of land cover and management effects (*P*-value < 0.05). Spatial map of (a) drought resistance $\alpha$ and (b) temperature sensitivity $\beta$ for EVI.

[Figure]

**Figure R20.** Ecosystem resistance to drought duration and temperature sensitivity selected for our analysis of land cover and management effects (*P*-value < 0.05). Spatial map of (a) drought resistance $\alpha$ and (b) temperature sensitivity $\beta$ for kNDVI.

**Reference**

D'Amato, A. W., Bradford, J. B., Fraver, S., and Palik, B. J.: Effects of thinning on drought vulnerability and climate response in north temperate forest ecosystems, Ecol Appl, 23, 1735–1742, https://doi.org/10.1890/13-0677.1, 2013.

Fawcett, D., Sitch, S., Ciais, P., Wigneron, J. P., Silva-Junior, C. H. L., Heinrich, V., Vancutsem, C., Achard, F., Bastos, A., Yang, H., Li, X., Albergel, C., Friedlingstein, P., and Aragão, L. E. O. C.: Declining Amazon biomass due to deforestation and subsequent degradation losses exceeding gains, Global Change Biology, n/a, https://doi.org/10.1111/gcb.16513, 2022.

Sanczuk, P., De Pauw, K., De Lombaerde, E., Luoto, M., Meeussen, C., Govaert, S., Vanneste, T., Depauw, L., Brunet, J., Cousins, S. A. O., Gasperini, C., Hedwall, P.-O., Iacopetti, G., Lenoir, J., Plue, J., Selvi, F., Spicher, F., Uria-Diez, J., Verheyen, K., Vangansbeke, P., and De Frenne, P.: Microclimate and forest density drive plant population dynamics under climate change, Nat Clim Change, 1–8, https://doi.org/10.1038/s41558-023-01744-y, 2023.

Silva Junior, C. H. L., Aragão, L. E. O. C., Anderson, L. O., Fonseca, M. G., Shimabukuro, Y. E., Vancutsem, C., Achard, F., Beuchle, R., Numata, I., Silva, C. A., Maeda, E. E., Longo, M., and Saatchi, S. S.: Persistent collapse of biomass in Amazonian forest edges following deforestation leads to unaccounted carbon losses, Sci Adv, 6, eaaz8360, https://doi.org/10.1126/sciadv.aaz8360, 2020.